# ReFeR: Improving Evaluation and Reasoning through Hierarchy of Models

**Yaswanth Narsupalli** *  
*IIT Kharagpur*

*yasshu.yaswanth@gmail.com*

**Abhranil Chandra** *  
*University of Waterloo*

*abhranil.chandra@gmail.com*

**Sreevatsa Muppirala** *  
*IIT Kharagpur*

*sreevatsa2002@gmail.com*

**Manish Gupta** †  
*Microsoft, India*

*gmanish@microsoft.com*

**Pawan Goyal** †  
*IIT Kharagpur*

*pawang@cse.iitkgp.ac.in*

**Reviewed on OpenReview:** *https://openreview.net/forum?id=otSHFe8wTf*

## Abstract

Assessing the quality of generative model outputs from large language models (LLMs) or vision-language models (VLMs), poses significant challenges. Traditional evaluation methods either rely on human assessment which is resource-intensive and not scalable or on automatic metrics that often correlate poorly with human preferences. Another approach is to train dedicated neural evaluators, but this typically requires substantial training data and compute. In this study, we thus introduce ReFeR, a tuning-free framework for evaluating generative outputs including both text and images, using a two-level hierarchy of pre-trained LLM and VLM evaluators. This multi-agent hierarchical strategy leverages additional compute at inference time by orchestrating multiple models and utilizing the increased test-time reasoning to boost performance. By having models themselves provide feedback and final judgments, ReFeR reduces the dependence on human evaluation. We rigorously evaluate ReFeR on four diverse evaluation benchmarks, where it surpasses prior methods in accuracy while also generating constructive feedback useful for downstream distillation and self-improvement via finetuning. Interestingly, ReFeR is also applicable for reasoning tasks - experiments on four reasoning benchmarks show ReFeR's superior collective reasoning abilities. We present two variants of the framework: ReFeR-Pro, optimized for accelerated performance, and ReFeR-Lite, offering a more test-time compute efficient solution. ReFeR-Lite is ∼12-14× more compute efficient than previous works while being comparably accurate to ReFeR-Pro. We make a PIP package, code and data publicly available[1],[2].

---

*Denotes Equal Contribution  
†Equal Supervision  
[1] https://github.com/yaswanth-iitkgp/ReFeR  
[2] https://pypi.org/project/refer-agents/

# 1 Introduction

The rapid production of content by LLMs and VLMs poses a challenge to traditional human-centric evaluation methods and conventional automatic metrics. Metrics like BLEU (Papineni et al., 2002), ROUGE (Lin, 2004), and METEOR (Banerjee & Lavie, 2005) for textual evaluation and CLIPScore (Hessel et al., 2022) for image to text evaluation, often misalign with human judgment and face limitations in assessing creative or nuanced responses. Recent studies suggest using LLMs as novel, reference-independent evaluators by assessing text quality based on predicted sequence likelihoods, bypassing the need for direct reference comparisons (Chen et al., 2023a). This has motivated researchers (Liu et al., 2023c; Chiang & Lee, 2023) to work on improving the evaluation capability of individual LLMs on text evaluation. Zhang et al. (2024) highlight that large models align more closely with human perceptual processes, thereby enhancing the evaluation of multimedia quality. Consequently, Chen et al. (2023b) leverage vision language models to provide explainable image quality evaluation by generating textual explanations, assessing fidelity, alignment, and aesthetics.

Surprisingly, despite the potential for improved performance by using ensembles of multiple vision-language models or large language models, there has been limited research on how to align evaluations from multiple VLMs or LLMs with human judgments. The recent advances in inference time compute methods have shown a promising direction to leverage additional compute generating more responses from the same or multiple models to improve performance in generative tasks (Wang et al., 2023; Yao et al., 2023). The prospect of thus taking inspiration from these developments and using multiple VLMs or LLMs together in a society of minds (Minsky, 1988) setup to solve the complex and nuanced problem of automatic and scalable evaluations is very promising. But it also introduces several uncertainties, including how to select the models, how many models to use, how to manage communication between different models and what prompting structure should be used to maximize the collective effect.

In this paper, we introduce Reason-Feedback-Review (ReFeR), a multi-agent framework drawing inspiration from the academic peer review process to enhance the evaluation of multimodal generative outputs like text generated by an LLM, an image generated by any model, or caption of an image generated by a VLM. By using multiple LLMs or VLMs as evaluators and feedback providers in a system akin to academic peer review, ReFeR enables a comprehensive evaluation of generative outputs across various domains, promoting model self-improvement, reasoning behind evaluation, and consistent score across runs. The paper outlines ReFeR's methodology, including its new prompting schema and the strategic use of LLMs or VLMs in roles parallel to peer reviewers and area chairs, facilitating a multi-dimensional evaluation through a hierarchical framework consisting of two levels: evaluation at the peer level and evaluation at the area chair level. ReFeR thus leverages increased computation at inference time by coordinating multiple models during evaluation to improve performance and by having models themselves produce feedback and final assessment, we introduce an automated model-based evaluation pipeline, reducing the dependence on human evaluators, thereby helping towards scalable oversight of generative model contents (Bowman et al., 2022; Kenton et al., 2024).

The framework is tested across two natural language generation (NLG) evaluation and two multimodal evaluation tasks. Interestingly, the ReFeR framework is generic enough to exhibit strong transfer to reasoning benchmarks as well. The multi-agent hierarchical setup, where multiple peer models deliberate, overseen by the area-chair model, promotes structured reasoning and reflective feedback, enabling complex reasoning and decision-making: we also test its reasoning ability on four reasoning benchmarks. This deliberative process also serves as a scalable generator of high-quality rich feedback-annotated synthetic instruction-tuning datasets. This imbues smaller models with evaluative capabilities more aligned with human preferences. This enables a scalable method to distill the automatic evaluation capabilities of ReFeR-like frameworks into smaller efficient models by fine-tuning them on these datasets, to achieve better correlation with human evaluation.

We present two variants of our proposed framework, ReFeR-Pro and ReFeR-Lite. ReFeR-Lite is $\sim14\times$ more test-time compute efficient than ReFeR-Pro. Both the variants outperform strong baselines on both text evaluation datasets: TopicalChat (Mehri & Eskenazi, 2020) and SummEval (Fabbri et al., 2021). ReFeR also beats baselines like Clipscore (Hessel et al., 2022), ImageReward (Xu et al., 2023a) and others on caption quality and image generation quality evaluation using ICQD (Levinboim et al., 2019) and AGIQA (Zhang et al., 2023) datasets respectively by large margins. Lastly, ReFeR also beats single agent methods (zero-shot CoT (Kojima et al., 2023), self correction (Huang et al., 2024)), and multi-agent methods like multi-agent

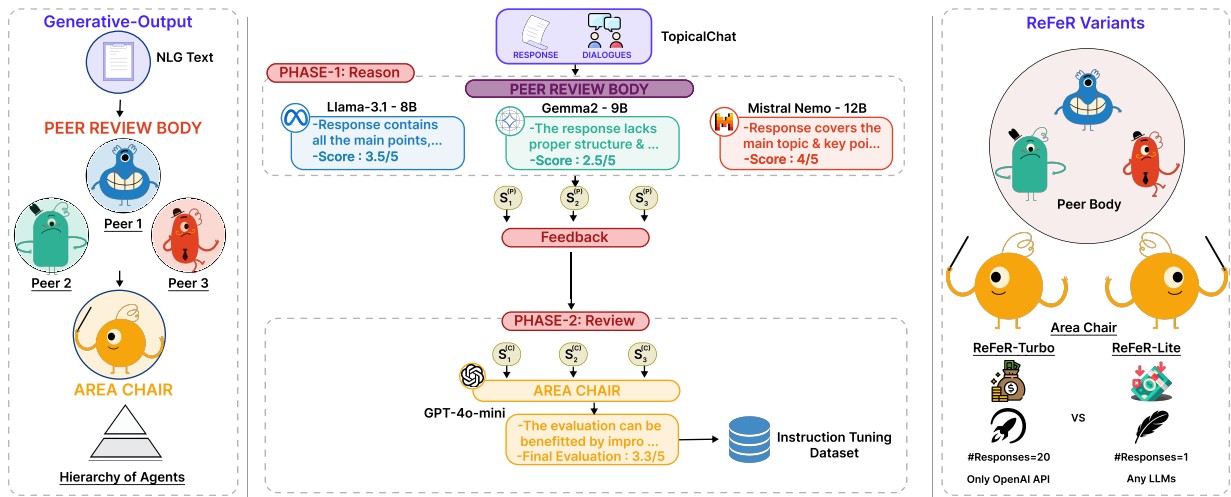

Figure 1: **Illustration of the ReFeR Framework on the TopicalChat dataset.** Refer to Fig. 5 (in the appendix) for illustration of ReFeR for multimodality and Algorithm 1 showing the framework's working. We use the predictions from AC to create an Instruction tuning dataset which can be used to improve the performance of smaller models as evaluators, shown in Appendix J

debate (Du et al., 2023) and multi-agent peer review (Xu et al., 2023b) on 3 out of 4 reasoning datasets, clearly outperforming on average while incurring lower costs than baselines.

To summarize, the primary contributions of our research are as follows: (1) Introduction of a general-purpose tuning-free multi-agent hierarchical framework leveraging inference time compute, called ReFeR, with its two variants, ReFeR-Pro and ReFeR-Lite. (2) We develop a novel prompting schema, with a novel evaluation guidelines component, specifically designed to improve the effectiveness of our framework both evaluation and reasoning tasks. (3) Empirical validation of the framework's evaluation and reasoning skills on four benchmarks each. (4) We conduct an in-depth analysis of our multi-agent framework, addressing key questions such as how to select models, how many models to use, and other critical aspects of model interactions.

## 2 Methodology

### 2.1 ReFeR Framework

Evaluating generative outputs in unverifiable domains without a predefined correct answer, such as assessing the quality of a research paper or open-ended responses, presents significant challenges. Inspired by the hierarchical peer review process in academia, we propose the ReFeR framework, which leverages a hierarchy of language models to systematically evaluate generative outputs. The ReFeR framework consists of two main modules, as depicted in Figure 1.

**1. Peer Review Body**

Let $G$ denote the generative output to be evaluated, $E_P$ represent the prompt for the peer and $E_{AC}$ represent the prompt for the area chair. Let $\mathcal{P} = \{P_1, P_2, \ldots, P_K\}$ be a set of $K$ peer agents, where each $P_i$ is a language model acting as a peer reviewer. Each peer agent independently evaluates $G$ according to $E_P$, producing a comment $C_i$ and a score $S_i \in \mathbb{R}$. This process is formalized as follows.

$$(C_i, S_i) = \text{Evaluate}_{P_i}(G, E_P), \quad \forall i \in \{1, 2, \ldots, K\} \tag{1}$$

**2. Area chair Evaluation**

An area chair agent $AC$, typically a larger or more capable language model, synthesizes the peer reviews to provide the final evaluation. The area chair considers the generative output $G$, the prompt $E_{AC}$, and the set of peer reviews $\{(C_i, S_i)\}_{i=1}^K$, producing a final comment $C_{\text{final}}$ and a final score $S_{\text{final}}$. $n$ is a hyperparameter

that denotes the number of responses for a given prompt.

$$\{(C_{\text{AC}}^{(j)}, S_{\text{AC}}^{(j)})\}_{j=1}^n \quad = \quad \left\{\text{Evaluate}_{AC}^{(j)}(G, E_{AC}, \{(C_i, S_i)\}_{i=1}^K)\right\} \tag{2}$$

$$S_{\text{final}} \quad = \quad \frac{1}{n}\sum_{j=1}^n S_{\text{AC}}^{(j)} \tag{3}$$

## 2.2   Prompting Schema

An essential aspect of assessing generative outputs with language model agents involves crafting prompts that elicit high-quality evaluations. Prior work, such as G-Eval by Liu et al. (2023c), introduced a structured evaluation schema, organizing the prompt into sections: task introduction, evaluation criteria, steps for evaluation, input presentation (context and target), and an evaluation form designed to output a numerical rating only. Subsequently, Chiang & Lee (2023) proposed an adjusted schema named Analyze-Rate, which prioritizes an analytical review followed by scoring, showing improved performance over G-Eval's prompt.

To further refine this approach, we introduce a new module in the evaluation schema called *Evaluation Guidelines* to enhance the model's understanding of the scoring criteria, akin to guidelines provided in traditional academic review processes. Evaluation guidelines can be automatically generated by prompting a language model with the prompt structure and some examples from the dataset and we call this process auto prompt. We give an example of this process in Appendix E showing the *Auto Prompt* for Engagingness prompt for TopicalChat. Alternatively, manually written human annotation guidelines of the dataset can be used. We also modified the evaluation form to include a critical comment or reasoning for the given score. The proposed evaluation schema is shown in Figure 4 in the Appendix.

## 2.3   ReFeR Variants

### 2.3.1   ReFeR-Pro

ReFeR-Pro leverages a hyperparameter $n$, representing the number of responses generated by the area chair agent. This variant generates multiple responses ($n = 20$) for each prompt, applying a scoring function that averages the scores across all generated responses, as described in Eq. 3.

The final comment $C_{\text{final}}$ is the list of all individual comments from the area chair evaluations. While ReFeR-Pro provides superior performance due to generating more evaluations per prompt, it incurs higher computational costs. Additionally, the use of $n = 20$ is often constrained to models from the OpenAI API, as other APIs supporting large models do not support this level of multiple response generation directly. Although it is possible to generate multiple responses by making repeated calls to the model (e.g., running the model 20 times with the same prompt), this approach is computationally expensive and less practical for large-scale evaluation tasks. This usage of the hyperparameter was first suggested by G-Eval and later used by Analyze-Rate.

### 2.3.2   ReFeR-Lite

To enhance flexibility and reduce computational overhead, we developed ReFeR-Lite, which removes the dependency on the parameter $n$ for the given performance. In this variant, only a single response ($n = 1$) is generated for each prompt, or $n$ is completely removed. This reduction in response generation is reflected in Eq 2, where $n$ is set to 1.

$$(C_{\text{final}}, S_{\text{final}}) = \text{Evaluate}_{AC}(G, E, \{(C_i, S_i)\}_{i=1}^K). \tag{4}$$

By relying on just one evaluation per prompt, ReFeR-Lite can be used with a wider variety of models, including open-source models, which do not natively support the generation of multiple responses with a single prompt. Despite generating fewer responses, ReFeR-Lite maintains competitive performance and offers significant cost savings. This makes it a more test-time computing-efficient solution for tasks where computational resources are limited or where evaluating large numbers of samples is required.

Both ReFeR-Pro and ReFeR-Lite use the same peer evaluation structure, but differ primarily in the area chair's response generation and model compatibility. ReFeR-Pro, with $n = 20$, offers potentially higher performance due to generating more evaluations but is restricted to models that support or can simulate

multiple response generation with a single prompt. In contrast, ReFeR-Lite provides greater flexibility and cost-efficiency by generating only a single response ($n = 1$) per prompt, making it more suitable for resource-constrained environments.

# 3 Experiments

## 3.1 Datasets

For NLG evaluation, we test our framework on SummEval (Fabbri et al., 2021) for summarization evaluation, and TopicalChat (Mehri & Eskenazi, 2020) for dialogue generation evaluation. For multimodal evaluation, we compare our framework on evaluating two types of task, image-to-text using ICQD (Image Caption Quality Dataset) (Levinboim et al., 2019) and text-to-image generation using AGIQA-1k by Zhang et al. (2023). For ICQD, we score model-generated captions and compare them with the average human annotated rating for the same. In AGIQA, we assess the quality of AI-generated images in reference to a given prompt and compare it with the mean opinion score (human annotations).

We also test our framework on 4 reasoning datasets: AQuA (Ling et al., 2017), BBH-DU (Srivastava et al., 2023), CSQA (Aggarwal et al., 2021) and GSM8k (Cobbe et al., 2021) which cover various reasoning tasks like Math, Commonsense and Date Understanding. Statistics and details about all the datasets are provided in Table 1. For more details about the datasets, refer to Appendix G. We test our framework on these reasoning tasks, where our framework answers a reasoning question with the label or numerical Answer after giving the reasoning. We calculate the accuracy of our answers in reference to the gold answers.

Table 1: **Dataset Statistics**. We list all the tasks we tackle in our paper and the datasets we used to show results with the number of samples used.[3]

| Dataset | Domain | Task | Samples | Answer | Scale |
|---------|--------|------|---------|--------|-------|
| TopicalChat | Dialogue Generation | NLG Evaluation | 360 | Rating (on 4 metrics) | 1-3 |
| SummEval | Summarization | NLG Evaluation | 1600 | Rating (on 4 metrics) | 1-5 |
| ICQD | Image-to-Text | Multimodal Evaluation | 864 | Caption Score | 0-100 |
| AGIQA | Text-to-Image | Multimodal Evaluation | 500 | Generation Score | 0-5 |
| AQuA | Math | Reasoning | 100 | Option | A-E |
| CSQA | Commonsense | Reasoning | 100 | Option | A-E |
| BBH-DU | Date Understanding | Reasoning | 100 | Option | A-F |
| GSM8k | Math | Reasoning | 100 | Number | - |

## 3.2 Baselines

### NLG Evaluation

While the current landscape of models for evaluating NLG responses includes reference-based methods such as BERTScore (Zhang et al., 2020a), UniEval (Zhong et al., 2022) and reference-free methods like GPTScore (Fu et al., 2023), we do not consider these models as baselines given they were clearly surpassed by G-Eval (Liu et al., 2023c) and later works (Chiang & Lee, 2023). Given our work primarily proposes a reference-free LLM-based evaluation for NLG, we do a comparative analysis primarily against G-Eval (Liu et al., 2023c) and Analyze-Rate (Chiang & Lee, 2023) only.

**G-Eval** performs evaluation by deploying a single LLM agent. This agent employs Auto-CoT (chain of thought) reasoning and a form-filling paradigm to ascertain the quality of NLG outputs, delivering only scores for the specific dimensions under scrutiny. They use 'n' hyperparameter to generate 20 responses and take the average score.

**Analyze-Rate** builds upon G-Eval, advocating for an enhanced prompt structure. This methodology incorporates a preliminary analysis phase before scoring, aiming to enrich the evaluative process for NLG tasks. Following G-Eval, they also consider the average score of 20 responses for each sample.

**LLM-as-Judge** (Zheng et al., 2023) is designed to evaluate LLMs and to rank them potentially creating ChatBOT Arena. We replicated the original setup for TopicalChat dataset.

---

[3]For Reasoning, a random subset of 100 was sampled from the original datasets, following (Chen et al., 2024a). 500 random samples were selected from the original AGIQA-1k to get a well-distributed dataset. We use 864 samples with usable image urls from the ICQD test dataset. We use the full test sets for the NLG Evaluation datasets.

**ChatEval** (Chan et al., 2023) proposed a single model multi-agent framework with varied persona to evaluate various NLG responses on open-ended questions and traditional NLP tasks by leveraging a debating structure among the agents. We reproduced their most optimal setup mentioned in the paper as baseline with 3 roles i.e. 3 agents and 2 discussion turns.

### MultiModal Evaluation

For multimodal evaluation, several works like HyperIQA (Su et al., 2020), DBCNN (Zhang et al., 2020b), IP-IQA (Qu et al., 2024) were proposed for image quality assessment, but all of these works are deep learning-based methods which leverage and depend on training a capable model. Hence we do not compare our framework against them directly.

**CLIP Score** (Hessel et al., 2022) evaluates how well an image aligns with a text description by using the CLIP model, which computes similarity scores between images and text embeddings.

**Image Reward** (Xu et al., 2023a) is a scoring model trained to assess the quality/alignment of generated images with text by comparing them against reference images using a reward model.

**Pick Score** (Kirstain et al., 2023) is another scoring model for the task of image text alignment, which is trained on human preference images 'picked' for a given text.

**X-IQE** (Chen et al., 2023b) leverages VLMs to evaluate text-to-image generation methods by generating textual explanations. We implement their Alignment dimension experiments to compare with our results on text-to-image generation dataset (AGIQA).

### Reasoning

We compare our framework against a variety of baseline methods across different categories. For single-agent methods, we select zero-shot Chain-of-Thought (CoT) and Self-Correct. For multi-agent frameworks, we compare against Multi-Agent Debate and Multi-Agent Peer Review, both of which use a single model acting as multiple agents.

**Zero-shot CoT** (Kojima et al., 2023) utilizes chain-of-thought prompting to generate reasoning processes and answers using a single agent.

**Self-Correct** (Huang et al., 2024) is a single-agent approach that enables an LLM to iteratively evaluate its own outputs, identify errors, and refine its responses through self-reflection.

**Multi-Agent Debate** (Du et al., 2023) involves a group of agents, where each agent observes the solutions provided by others, updates its own solution accordingly, and repeats this process through multiple iterations.

**Multi-Agent Peer Review** (Xu et al., 2023b) is a multi-agent system in which each agent independently generates a solution, reviews the solutions of others, and assigns confidence scores to its reviews. Agents then revise their initial solutions based on the received peer reviews. This revision is repeated through multiple iterations/rounds of peer review. We used the default number of rounds (3) mentioned by the authors.

### 3.3   Implementation Details

**NLG Evaluation:** Our framework for NLG evaluation employs Llama-3.1-8B-Instruct (Meta-AI, 2024), Mistral-Nemo-12B (Mistral-AI, 2024) and Gemma-2-9B (Google-Research, 2024) as the peer models and GPT-4o-mini (OpenAI, 2024b) as the area chair model. We use Together-AI (2023)'s API for the peer models, but since these are small open-source models, they can also be deployed locally. For the baselines, we follow the original setups (with GPT-4o-mini) proposed by Liu et al. (2023c) and Chiang & Lee (2023). As mentioned in Section 2.3, we vary the hyperparameter $n$ for the two ReFeR variants. For more details on other hyperparameters, refer to Appendix C.

**Multimodal Evaluation:** For multimodal evaluation, our framework uses only 2 peers: Gemini-1.5-Flash (DeepMind, 2024) and GPT-4o-mini (OpenAI, 2024b). We use GPT-4o (OpenAI, 2024a) as the area chair model. We choose only 2 peers for multimodal evaluation setup considering the cost and availability of VLMs of similar strength. More details on the number of peers and how to choose peers are described in Section 5. The baselines like CLIPScore (Hessel et al., 2022), ImageReward (Xu et al., 2023a), PickScore (Kirstain et al., 2023) are implemented following the codes provided in their official repositories.

**Reasoning:** We use the same setup as our NLG evaluation for all our reasoning experiments following similar prompting structure except using evaluation guidelines which is irrelevant in reasoning tasks. All the

baselines were implemented and evaluated using the scripts provided by Xu et al. (2023b) in their official repository.

## 4 Results and Discussions

This section presents the experimental results evaluating ReFeR's effectiveness in assessing text, multimodal outputs, and reasoning capabilities. Experimental details are provided in Section 3.3, hyperparameters in Appendix C, and prompts in Appendix R.

### 4.1 NLG Evaluation

We evaluate ReFeR's performance on two datasets: TopicalChat and SummEval. For TopicalChat, we assess dialog system responses based on four metrics: Coherence, Engagingness, Groundedness, and Naturalness. For SummEval, we evaluate article summaries using Coherence, Consistency, Fluency, and Relevance metrics. Following (Liu et al., 2023c) and (Chiang & Lee, 2023), we compare the generated scores with human-annotated ground truth using Spearman ($\rho$) and Kendall-tau ($\tau$) correlations. Results for TopicalChat are presented in Table 2, with SummEval results in Appendix F. All results are averaged over three runs. The table first shows individual peer performances using our peer prompt, followed by baselines Analyze-Rate (Chiang & Lee, 2023) and G-Eval (Liu et al., 2023c). ReFeR Pro outperforms all baselines on most metrics and excels on average. ReFeR Lite, our cost-effective model, ranks second on average despite generating a single response instead of 20 like G-Eval and Analyze-Rate. G-Eval sometimes outperforms Analyze-Rate despite only generating scores, while both Analyze-Rate and ReFeR provide analysis in addition to scores, offering the potential for model improvement. The key findings from this experiment are: (1) Both ReFeR Pro and ReFeR Lite outperform baselines. (2) ReFeR Lite with $n = 1$ also achieves comparable performance while being significantly cheaper.

While generating multiple responses (e.g., $n = 20$ as in G-Eval) is theoretically possible with any LLM, it poses substantial practical challenges. For instance, evaluating the TopicalChat dataset (360 samples, 4 metrics) would require approximately 28,800 model calls with an average input token size of 675 for TopicalChat. This approach becomes impractical in terms of cost, time, and computational resources, especially for models without the throughput ($n = 20$) capabilities of the OpenAI API. Hence ReFeR-Lite can be an option in such cases.

Table 2: **Comparison of ReFeR with baselines for NLG evaluation on the TopicalChat dataset (all baselines and ReFeR uses GPT-4o-mini as the backbone).** Results are averaged across 3 runs. The best results are bolded, and the second-best are underlined. *Costs for ReFeR Pro and ReFeR Lite include only AC API cost, as open-source peer models can be deployed locally and so do not involve API costs. Peer model costs based on API pricing from services like (Together-AI, 2023) are also provided for reference. Relative costs are shown as fractions of the most expensive method. GPT-4o-mini row denotes the performance of the AC model with the peer setup, Peer Average row denotes the correlation when the average of the 3 peer scores is considered.

| | Method | Coherence | | Engagingness | | Groundedness | | Naturalness | | Average | | Cost |
|---|---|---|---|---|---|---|---|---|---|---|---|---|
| | | $\rho$ | $\tau$ | $\rho$ | $\tau$ | $\rho$ | $\tau$ | $\rho$ | $\tau$ | $\rho$ | $\tau$ | (Relative) |
| Peer Agents | Llama-3.1-8B | 0.380 | 0.324 | 0.400 | 0.342 | 0.444 | 0.414 | 0.320 | 0.268 | 0.386 | 0.337 | 0.13 |
| | Mistral Nemo-12B | 0.409 | 0.346 | 0.594 | 0.501 | 0.442 | 0.414 | 0.411 | 0.348 | 0.464 | 0.402 | 0.23 |
| | Gemma-2-9B | 0.536 | 0.453 | 0.615 | 0.527 | 0.582 | 0.545 | 0.519 | 0.430 | 0.563 | 0.489 | 0.20 |
| | GPT-4o-mini | 0.518 | 0.438 | 0.618 | 0.527 | 0.589 | 0.549 | 0.540 | 0.457 | 0.566 | 0.493 | 0.13 |
| | Peer Average | 0.547 | 0.433 | 0.648 | 0.519 | 0.577 | 0.510 | 0.512 | 0.396 | 0.539 | 0.447 | 0.56 |
| Baselines | Analyze-Rate | 0.551 | 0.465 | 0.638 | 0.544 | 0.615 | 0.569 | 0.562 | 0.476 | 0.591 | 0.514 | 0.77 |
| | G-Eval | 0.581 | **0.493** | 0.636 | **0.546** | 0.593 | 0.555 | 0.558 | 0.470 | 0.592 | **0.516** | 0.13 |
| | LLM-as-Judge | 0.510 | 0.445 | 0.593 | 0.519 | 0.556 | 0.467 | 0.534 | 0.471 | 0.548 | 0.476 | 0.13 |
| | ChatEval | 0.551 | 0.471 | 0.624 | 0.538 | 0.522 | 0.428 | 0.557 | **0.478** | 0.564 | 0.479 | 0.78 |
| Ours | ReFeR Pro | **0.592** | 0.458 | **0.677** | 0.536 | **0.645** | **0.588** | **0.616** | 0.473 | **0.632** | 0.514 | 1.0* |
| | ReFeR Lite | 0.561 | 0.479 | 0.636 | 0.543 | 0.618 | 0.575 | 0.591 | 0.416 | 0.602 | 0.503 | 0.13* |

### 4.2 Multimodal Evaluation

To assess the multimodal applicability of ReFeR, we conducted experiments on two tasks: image generation quality evaluation using the AGIQA dataset (text-to-image setting) and image caption evaluation using

the ICQD dataset (image-to-text setting). Table 3 presents the results of these experiments. Following previous deep learning-based works such as (Zhang et al., 2023), we report Spearman's $\rho$ and Kendall's $\tau$ rank correlations. Key findings include the following.

- ICQD dataset: Both variants of ReFeR outperform all baselines. Notably, although individual peers show low correlations, AC effectively countered this, resulting in better correlation.
- AGIQA dataset: ReFeR Pro outperforms all baselines, while ReFeR Lite outperforms ClipScore and X-IQE but falls short of ImageReward and PickScore.

We attribute the performance difference in the AGIQA dataset to the fact that both ImageReward and PickScore involve training based on human preferences, which may have contributed to their superior performance compared to our ReFeR Lite variant. But, our ReFeR-Lite has clearly surpassed a single VLM based method X-IQE by a large margin showing the effectiveness of the framework.

Table 3: **Multimodal Evaluation Results.** Comparison of caption quality and image generation quality score correlations with human scores on ICQD and AGIQA datasets, respectively. *X-IQE is a text-to-image VLM-based method, so we don't show it for Caption Quality.

|  | Method | Caption Quality | | Image Quality | | Cost |
|---|---|---|---|---|---|---|
|  |  | $\rho$ | $\tau$ | $\rho$ | $\tau$ | (Relative) |
| Peer Agents | Gemini-1.5-Flash | 0.135 | 0.098 | 0.341 | 0.268 | 0.07 |
|  | GPT-4o-mini | 0.200 | 0.145 | 0.502 | 0.392 | 0.01 |
| Baselines | Clip Score | 0.310 | 0.233 | 0.522 | 0.366 | - |
|  | ImageReward | 0.433 | 0.302 | 0.634 | 0.451 | - |
|  | Pick Score | 0.352 | 0.241 | 0.627 | 0.442 | - |
|  | X-IQE* | - | - | 0.410 | 0.307 | 0.05 |
| Ours | ReFeR Pro | **0.497** | **0.347** | **0.657** | **0.467** | 1.0 |
|  | ReFeR Lite | 0.459 | 0.336 | 0.599 | 0.442 | 0.14 |

### 4.3 Reasoning

We hypothesize that our framework enhances the overall reasoning capabilities of area chair by utilizing multiple models collaboratively, leading to improved decision-making. To verify this, we compare ReFeR's reasoning capabilities against other frameworks, including zero-shot-CoT, single-agent frameworks, and same-model multi-agent frameworks. Table 4 presents the results of these experiments, with all results averaged across 3 runs, following the setup in (Chen et al., 2024a). Key observations:

- On average, ReFeR outperforms all other baselines across the tested benchmarks.
- In the BBH Date Understanding benchmark, debating-type frameworks like Multi-Agent Debate show better results than ReFeR. This may be attributed to the nature of the benchmark, which involves understanding dates and resolving conflicts. Such tasks benefit from inter-agent discussions, which are possible in a debating setup but not in ReFeR's hierarchical framework.
- ReFeR outperforms baselines on the AQuA benchmark because the hierarchical structure allows the area chair to synthesize peer inputs efficiently, avoiding confusion. In contrast, debate formats may cause models to introduce conflicting reasoning, which is less effective for tasks requiring precise reasoning like AQuA.
- Considering overall cost and performance, both variants of ReFeR demonstrate significant advantages in terms of cost-efficiency compared to corresponding multi-agent models.
- Further analysis on response quality between ReFeR and other methods on GSM8k dataset using reference rationale as premise we get a HHEM score (Bao et al., 2024) of 0.33 whereas other baselines have a score ranging from 0.102 to 0.115. Higher HHEM score indicates better response quality. More details on the analysis are provided in the Appendix-O.

## 5  Analysis

In this section, we perform an analysis of the framework to understand the impact of different components and choices.

Table 4: **Experimental results on Reasoning tasks.** Comparison of ReFeR performance (accuracy) with single-agent and multi-agent method baselines. All results are averaged across 3 runs. Cost*- Costs are shown as relative to the most expensive method.

| Method Type | Methods | AQuA | BBH_DU | CSQA | GSM8k | Average | Cost* |
|---|---|---|---|---|---|---|---|
| Peer Agents | Llama-3.1-8B | $26.3 \pm 5.1$ | $28.0 \pm 7.8$ | $68.3 \pm 4.0$ | $40.0 \pm 11.4$ | $40.7 \pm 7.1$ | 0.03 |
| | Mistral Nemo-12B | $43.0 \pm 3.6$ | $55.7 \pm 4.6$ | $65.7 \pm 6.1$ | $54.7 \pm 11.5$ | $54.8 \pm 6.5$ | 0.05 |
| | Gemma-2-9B | $50.7 \pm 2.3$ | $70.3 \pm 6.5$ | $75.7 \pm 4.5$ | $79.3 \pm 4.0$ | $69.0 \pm 4.3$ | 0.04 |
| Single Agent | zero-shot-CoT | $60.7 \pm 1.5$ | $88.0 \pm 1.7$ | $76.0 \pm 1.7$ | $95.3 \pm 1.2$ | $80.0 \pm 1.5$ | 0.06 |
| | Self Correction | $78.7 \pm 0.6$ | $\underline{92.7} \pm 0.6$ | $54.0 \pm 1.0$ | $93.3 \pm 1.5$ | $79.7 \pm 0.9$ | 0.20 |
| Multi-Agent | Multi-Agent Debate | $54.0 \pm 3.5$ | $\mathbf{94.3} \pm 0.6$ | $\underline{79.0} \pm 1.7$ | $\underline{95.7} \pm 0.6$ | $80.8 \pm 1.6$ | 0.97 |
| | Multi-Agent Peer Review | $62.0 \pm 4.4$ | $91.7 \pm 0.6$ | $77.0 \pm 1.0$ | $95.3 \pm 1.5$ | $81.5 \pm 1.9$ | 1.0 |
| Ours | ReFeR Pro | $\mathbf{85.0} \pm 1.0$ | $92.0 \pm 2.0$ | $\mathbf{79.3} \pm 1.2$ | $\mathbf{96.0} \pm 0.0$ | $\mathbf{88.1} \pm 1.0$ | 0.93 |
| | ReFeR Lite | $\underline{81.0} \pm 2.0$ | $91.0 \pm 2.0$ | $\mathbf{79.3} \pm 1.2$ | $93.3 \pm 2.1$ | $\underline{86.2} \pm 1.8$ | 0.18 |

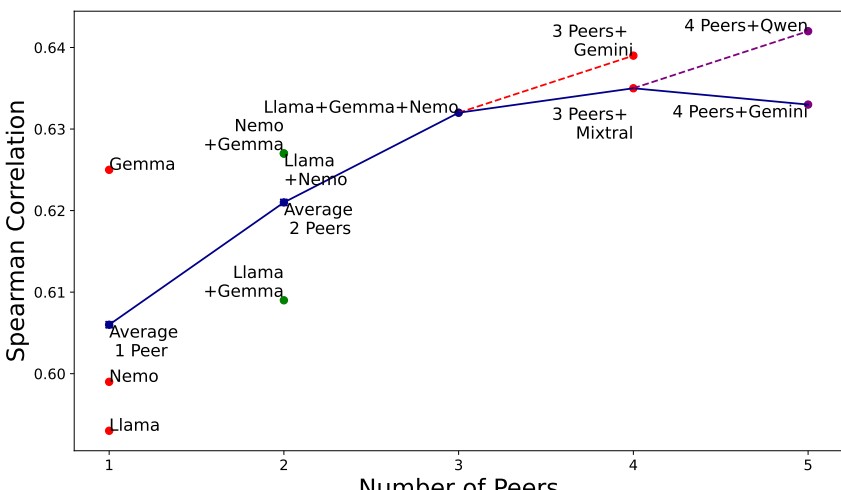

Figure 2: **Framework Ablation.** Results obtained on ReFeR-Pro by progressively adding different peers for the TopicalChat Dataset. The points in the figure indicate the performance of ReFeR when specific labelled peers were used in conjunction with the area chair (GPT-4o-mini). "3 Peers" refers to the Llama, Nemo, and Gemma models being used as peers. "4 Peers" includes the same 3 peers along with the Mixtral model added as fourth peer. Detailed results are presented in Table 6.

## 5.1 Peer Ablation

To evaluate the impact of number of peer agents and composition on ReFeR's performance, we conducted a peer ablation study using the TopicalChat dataset shown in Fig. 2.

Our findings indicate that increasing the number of peers generally improves the framework's overall correlation, as evidenced by the main branch in Fig. 2. We experimented with varying peer combinations and numbers to distinguish between the effects of adding another peer versus a better-performing peer. Due to the impracticality of exploring all possible combinations with six peers, we selected a subset based on individual performances, costs, and model sizes.

Fig. 2 demonstrates that while the framework's average performance generally increases with more peers, adding a relatively weaker model can result in better performance than the base (1 peer) but not necessarily the highest overall. For instance, with five peers, the combination of four peers plus Qwen yields the best performance, whereas four peers plus Gemini (weaker at this task) performs closer to the three-peer configuration. Notably, the performance gain from four or five peers compared to three peers is not substantial. This observation suggests that using three peers may be an optimal choice, balancing performance improvements with computational efficiency.

## 5.2 Selecting Peers and area chair

For optimal peer selection, we recommend using a group of capable peers chosen based on their individual performances in performance assessment. After assessing individual performances, top-performing peers can

be selected considering both cost and performance. As shown in Table 6, Gemma2-9B is the top performer across all metrics, while Mistral Nemo 12B and Llama-3.1 8B offer comparable performance at lower costs. Consequently, we selected Gemma2-9B, Llama-3.1 8B, and Mistral Nemo 12B as our peers. This selection is also crucial for enabling local GPU deployment of the peers.

To understand the framework's effectiveness under various conditions, we conducted a study by fixing the peers and changing the area chair. Particularly, we choose an area chair which is relatively weaker than not just GPT-4o-mini but also our best peer, Gemma2-9B, at this task. Hence, we choose Qwen1.5 - 72B. Table 5 presents the results using the ReFeR Lite setting on the TopicalChat dataset. We observed that although we used Qwen as AC (whose individual performance is less than the best peer), we get improved performance compared to the respective individual performance. But we see that one of the peer's (Gemma2-9B) has a correlation of 0.568 hence we deduce that if the AC model is relatively stronger than most of the peers then we get improved performance but to get the best results out of the framework we see that we need a larger or better model as AC to better utilize the evaluations done by the peers and incorporate them in it's own evaluation.

Table 5: **Results on TopicalChat using the open-source model Qwen1.5-72B as the area chair.** We were unable to include results for ReFeR Pro with Qwen as the area chair due to the limitation of not being able to use n=20.

| Method | Coherence | | Engagingness | | Groundedness | | Naturalness | | Avg | |
|---|---|---|---|---|---|---|---|---|---|---|
| | $\rho$ | $\tau$ | $\rho$ | $\tau$ | $\rho$ | $\tau$ | $\rho$ | $\tau$ | $\rho$ | $\tau$ |
| (Peer) Llama-3.1-8B | 0.417 | 0.357 | 0.418 | 0.357 | 0.488 | 0.455 | 0.346 | 0.289 | 0.417 | 0.365 |
| (Peer) Mistral Nemo-12B | 0.416 | 0.352 | 0.567 | 0.475 | 0.453 | 0.424 | 0.396 | 0.339 | 0.458 | 0.397 |
| (Peer) Gemma-2-9B | **0.549** | **0.465** | **0.623** | **0.534** | 0.583 | 0.545 | 0.520 | 0.431 | **0.568** | **0.494** |
| Qwen (Individual Performance) | 0.465 | 0.399 | 0.524 | 0.459 | 0.471 | 0.441 | 0.508 | 0.437 | 0.492 | 0.434 |
| ReFeR Lite (Qwen) | 0.496 | 0.422 | 0.609 | 0.522 | **0.587** | **0.550** | **0.527** | **0.450** | 0.555 | 0.486 |

This observation aligns with the original analogy of research paper peer review, where the area chair is typically a senior researcher with a potentially better understanding than most peer reviewers, thus being given more importance or final judgment authority. In cases where performance assessment is not feasible to determine the most suitable models, the LLM Leaderboard on HuggingFace (2024) can be consulted to select appropriate models based on the specific task requirements, cost considerations, GPU availability, and time constraints.

Table 6 shows the individual performances of the 6 open-source models we chose and then the ablation of an increasing number of peers and the ablation of adding different models. The models we used are Llama-3.1-8B, Mistral-Nemo-12B, Gemma-2-9B, Mixtral-8x7B, Gemini-1.5-Pro, Qwen-1.5-72B.

We have also performed AC ablation experiments with another AC model to show the difference of performance based on the model chosen. We have chosen our best peer model Gemma2-9B as the AC model while using the same peer models and their responses, as our original setup. We use the same hyperparameters for the AC model(Gemma) as our original setup. We also show the original GPT-4o-mini results to compare.

From the results of Table 7, we can see that the peer performance of Gemma2-9B is relatively lesser than that of GPT-4o-mini. This same trend is observed when using these models as AC in the ReFeR framework. The better performing model GPT-4o-mini gives better correlation as the AC too. This further supports our statements in Section 5.2, where we show that a stronger model has to be chosen as an AC to get the best performance from the framework.

### 5.3 Error Analysis

To assess the framework's effectiveness in both evaluation and reasoning tasks, we conducted an error analysis, with results shown in Fig. 3. In this analysis, a TopicalChat sample's evaluation score for each metric is considered correct if it falls within a given threshold range of 25%. In the TopicalChat dataset evaluation, the area chair provided correct scores 42.6% of the time when one or two peers provided a correct answer, demonstrating the AC's ability to leverage partially correct peer scores effectively. The AC made mistakes only 11.9% of the time when at least one peer was correct. However, the AC was correct only in 2% of the cases where all the peers were incorrect, suggesting that the AC may require at least one correct peer input to avoid confusion and give a correct score. For reasoning tasks, the AC was incorrect for only 2.9% of cases where atleast one peer is correct, showing similiar observation as evaluation. And the AC was

Table 6: **Ablation analysis of the effect of different models used.** Comparison of ReFeR Pro results on TopicalChat with Different Peer Configurations. The method column shows what peers were used with the AreaChair (GPT-4o-mini). 4 Peers in the last rows denotes the 3 peers (Llama, Nemo, Gemma) and Mixtral. *These rows show the individual performance of the peers, not the framework's performance when the peer is used.

| | Peers Used | Coherence | | Engagingness | | Groundedness | | Naturalness | | Average | |
|---|---|---|---|---|---|---|---|---|---|---|---|
| | | $\rho$ | $\tau$ | $\rho$ | $\tau$ | $\rho$ | $\tau$ | $\rho$ | $\tau$ | $\rho$ | $\tau$ |
| Individual results* | Llama | 0.417 | 0.357 | 0.418 | 0.357 | 0.488 | 0.455 | 0.346 | 0.289 | 0.417 | 0.365 |
| | Nemo | 0.416 | 0.352 | 0.567 | 0.475 | 0.453 | 0.424 | 0.396 | 0.339 | 0.458 | 0.397 |
| | Gemma | 0.549 | 0.465 | 0.623 | 0.534 | 0.583 | 0.545 | 0.520 | 0.431 | 0.568 | 0.494 |
| | Mixtral | 0.440 | 0.373 | 0.552 | 0.467 | 0.491 | 0.458 | 0.469 | 0.390 | 0.488 | 0.422 |
| | Gemini | 0.352 | 0.300 | 0.460 | 0.387 | 0.498 | 0.466 | 0.419 | 0.352 | 0.432 | 0.376 |
| | Qwen | 0.465 | 0.399 | 0.524 | 0.459 | 0.471 | 0.441 | 0.508 | 0.437 | 0.492 | 0.434 |
| | Average | 0.440 | 0.374 | 0.524 | 0.446 | 0.497 | 0.465 | 0.443 | 0.373 | 0.476 | 0.415 |
| 1 Peer | Llama | 0.542 | 0.423 | 0.603 | 0.479 | 0.628 | 0.556 | 0.599 | 0.460 | 0.593 | 0.479 |
| | Nemo | 0.558 | 0.440 | 0.684 | 0.548 | 0.617 | 0.555 | 0.539 | 0.414 | 0.599 | 0.489 |
| | Gemma | 0.564 | 0.448 | 0.680 | 0.552 | 0.635 | 0.578 | 0.622 | 0.481 | 0.625 | 0.515 |
| | Average | 0.555 | 0.437 | 0.656 | 0.526 | 0.626 | 0.563 | 0.587 | 0.452 | 0.606 | 0.494 |
| 2 Peers | Llama+Gemma | 0.565 | 0.440 | 0.656 | 0.524 | 0.593 | 0.535 | 0.621 | 0.481 | 0.609 | 0.495 |
| | Llama+Nemo | 0.577 | 0.450 | 0.692 | 0.553 | 0.621 | 0.570 | 0.621 | 0.480 | 0.627 | 0.513 |
| | Nemo+Gemma | 0.567 | 0.443 | 0.685 | 0.547 | 0.622 | 0.573 | 0.632 | 0.490 | 0.627 | 0.513 |
| | Average | 0.570 | 0.444 | 0.677 | 0.541 | 0.612 | 0.559 | 0.624 | 0.484 | 0.621 | 0.507 |
| 3 Peers | Llama+Gemma+Nemo | 0.589 | 0.458 | 0.689 | 0.550 | 0.623 | 0.574 | 0.626 | 0.486 | 0.632 | 0.517 |
| 4 Peers | 3 Peers + Mixtral | 0.596 | 0.463 | 0.682 | 0.541 | 0.629 | 0.572 | 0.634 | 0.494 | 0.635 | 0.517 |
| | 3 Peers + Gemini | 0.601 | 0.469 | 0.688 | 0.550 | 0.644 | 0.590 | 0.623 | 0.485 | 0.639 | 0.523 |
| 5 Peers | 4 Peers + Gemini | 0.584 | 0.455 | 0.686 | 0.545 | 0.623 | 0.572 | 0.640 | 0.495 | 0.633 | 0.517 |
| | 4 Peers + Qwen | 0.601 | 0.467 | 0.682 | 0.545 | 0.646 | 0.592 | 0.637 | 0.498 | 0.642 | 0.526 |

Table 7: **Choosing model for AC**: Below Table shows individual performance of AC model and model acting as AC in the ReFeR Framework. The purpose of this table is to understand correlation of capability of the AC and our framework. We See that if gpt-4o-mini is better than gemma in base performance then gpt-4o-mini is better at being AC than gemma

| Method | Coherence | | Engagingness | | Groundedness | | Naturalness | | Average | |
|---|---|---|---|---|---|---|---|---|---|---|---|
| | $\rho$ | $\tau$ | $\rho$ | $\tau$ | $\rho$ | $\tau$ | $\rho$ | $\tau$ | $\rho$ | $\tau$ |
| Gemma-2-9B (Peer Setup) | 0.536 | 0.453 | 0.615 | 0.527 | 0.582 | 0.545 | 0.519 | 0.430 | 0.563 | 0.489 |
| Gemma-2-9B (n=20) | 0.556 | 0.466 | 0.617 | 0.521 | 0.577 | 0.538 | 0.530 | 0.434 | 0.570 | 0.490 |
| ReFeR-Pro (Gemma AC) | 0.569 | 0.440 | **0.684** | 0.540 | **0.643** | **0.581** | 0.590 | 0.451 | 0.621 | 0.503 |
| ReFeR-Lite (Gemma AC) | 0.552 | 0.463 | 0.624 | 0.533 | 0.607 | 0.567 | 0.574 | 0.484 | 0.589 | 0.512 |
| GPT-4o-mini (Peer Setup) | 0.518 | 0.438 | 0.618 | 0.527 | 0.589 | 0.549 | 0.540 | 0.457 | 0.566 | 0.493 |
| ReFeR-Pro (gpt-4o-mini as AC) | **0.585** | 0.454 | 0.673 | 0.535 | 0.628 | 0.577 | **0.625** | 0.484 | **0.628** | 0.513 |
| ReFeR-Lite (gpt-4o-mini as AC) | 0.552 | **0.467** | 0.640 | **0.550** | 0.596 | 0.558 | 0.599 | **0.505** | 0.597 | **0.520** |

correct 14% of the time, even when all peers were incorrect, indicating a better ability to disregard clearly incorrect answers from the smaller peer models. This suggests that the AC, when using reasoning tasks, may not always rely on peers and can function independently in such cases.

More analysis about prompt optimization, communication strategies, inference and test-time compute, and statistical significance tests can be found in Appendices D, K, L and M, respectively.

# 6 Related Work

**NLG & Multimodal Evaluation:** Recent advancements in NLG evaluation include GPTScore (Fu et al., 2023), which uses generative pre-training models to assess text quality, and G-Eval (Liu et al., 2023c), employing a chain-of-thoughts approach with form-filling methodology. Chiang & Lee (2023) highlighted limitations in G-Eval's automated CoT alignment with human evaluations. TIGERScore (Jiang et al., 2023) offers detailed error analysis using fine-tuned Llama-2, while FusionEval (Shu et al., 2024) integrates auxiliary evaluators with a primary LLM for scoring. X-Eval (Liu et al., 2023a) introduces a two-stage instruction tuning framework for diverse evaluation dimensions. ChatEval (Chan et al., 2023) proposes a multi-agent referee system using autonomous debating among agents with different personas to evaluate response quality. While similar to our approach, it primarily relies on debate methodology using the same models under varied personas, whereas our method employs diverse models as peers and area chairs with a richer prompting schema. Mixture of Agents (MoA) (Wang et al., 2024b) is another work which explores the how to harness the collective expertise of multiple LLMs. In multimodal evaluation, CLIP Score (Hessel et al., 2022), Image-Reward (Xu et al., 2023a), and Pick Score (Kirstain et al., 2023) assess image-text

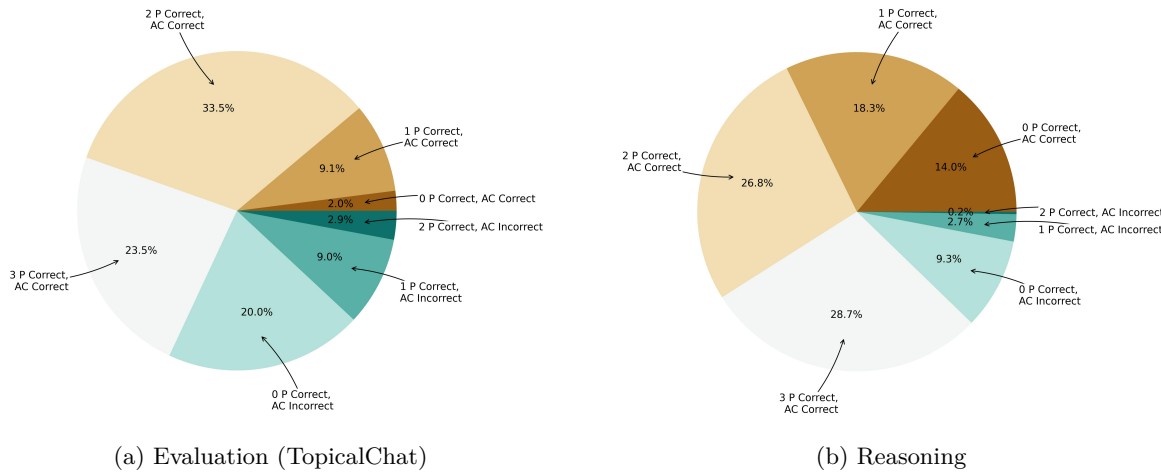

(a) Evaluation (TopicalChat)         (b) Reasoning

Figure 3: **Performance analysis wrt framework scale.** Pie-charts showing Peer and AC performance on evaluation and reasoning tasks. (P- Peer model, AC- area chair Model)

alignment using pre-trained models. Deep learning methods like HyperIQA (Su et al., 2020) and IP-IQA (Qu et al., 2024) have shown improvements in this domain. Later, X-IQE (Chen et al., 2023b) introduced using VLMs for the task of image-quality assessment.

**Reasoning using LLMs:** Single-agent methods like Zero-shot CoT (Kojima et al., 2023) have improved language models' reasoning capabilities using Chain-of-Thought prompting. Self-correction (Huang et al., 2024) mimics human self-reflection to address reasoning errors. In multi-agent frameworks, Du et al. (2023) introduced a same-model approach using peer solutions for individual improvement, while Pham et al. (2023) proposed embedding-based communication to optimize reasoning. Xu et al. (2023b) developed a framework inspired by academic peer review, emphasizing iterative improvement through peer feedback. This differs from our method, which involves an area chair reviewing peer responses without direct inter-peer communication. ReConcile (Chen et al., 2024a) structured a multi-model, multi-agent framework as a round table conference, demonstrating enhanced reasoning through discussion and consensus. We expand on why we did not consider ReConcile as a baseline in Appendix H. Wang et al. (2024c) proposed selecting the most coherent response from multiple reasoning chains, offering an alternative approach to consensus-building and improving reasoning accuracy. For more recent related works discussion, please refer to Appendix I.

## 7 Conclusion

In this work, we propose ReFeR (**Re**ason-**Fe**edback-**R**eview), a hierarchical model framework that utilizes smaller yet capable models as peers and a powerful model as the area chair. The area chair leverages the reasoning and feedback from the peer models to produce a final review for evaluating given images or texts. We demonstrate ReFeR's efficacy across two NLG evaluation tasks, two multimodal evaluation tasks, and four reasoning tasks, outperforming various baselines while maintaining favorable cost-performance trade-offs. We also present two variants: ReFeR-Pro and ReFeR-Lite. Notably, our Lite version achieves performance comparable to other works and to ReFeR-Pro, while being significantly more efficient in terms of compute used.

Our framework, while robust in many aspects, has some limitations. One notable constraint is the potential computational cost when using large models as both peers and area chairs, especially in resource-limited environments. Additionally, the framework currently lacks an interactive discussion phase between peer models, which could further improve collective reasoning. In some scenarios, such as when a weaker model is used as the area chair, the performance may not be optimal. Lastly, while our framework has shown promising results on text and image evaluation tasks, it remains untested in other modalities, which could present unique challenges in scaling and complexity and is an interesting area for future research.

**Broader Impact Statement**

This work adheres to the TMLR Ethics Guidelines, ensuring that all evaluations and methodologies applied in the ReFeR framework were conducted with fairness, transparency, and integrity. Since ReFeR operates as a framework for evaluating machine-generated content, the primary ethical concerns are related to ensuring unbiased assessments and avoiding unintended model biases in evaluations. We carefully selected models to minimize potential biases, but the limitations of the models used could still introduce unintended biases, which we will continue to address in future improvements. No human subjects were involved in the experiments conducted for this study. Additionally, we commit to making our code and datasets available for further research, study and improvements.

To ensure the reproducibility of our results, we provide a detailed description of the ReFeR framework, including the structure of the hierarchical evaluation system and its variants. All hyperparameters, evaluation criteria, and the models used are described in the main text and appendices. The datasets utilized for evaluation and reasoning tasks are publicly available, as mentioned in Section 3.1. Additionally, we will release the source code, along with instructions for running the experiments, on an anonymous repository. Clear explanations for the model selection process, evaluation metrics, and experimental setups are also included to facilitate replication by other researchers.

# 8 Acknowledgments

We would really like to thank Microsoft Research for their Accelerating Foundation Models Research initiative for generously funding our research and helping us with the GPUs and API credits. We would also like to thank Ganesh Jawahar for helping us with brainstorming the problem statement initially.

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

# A  Future Works

Future research can explore incorporating additional elements from the academic peer review process, such as the author discussion phase, to simulate a more interactive review environment. Expanding the framework to include evaluations beyond text and images, such as video and audio content, could further enhance its applicability. Another promising direction is to develop various communication strategies between peers and the area chair to optimize evaluation and feedback cycles. Moreover, experimenting with different numbers of area chairs of varying strength could help in understanding the impact of multiple, potentially conflicting, judgments on the final evaluation outcomes.

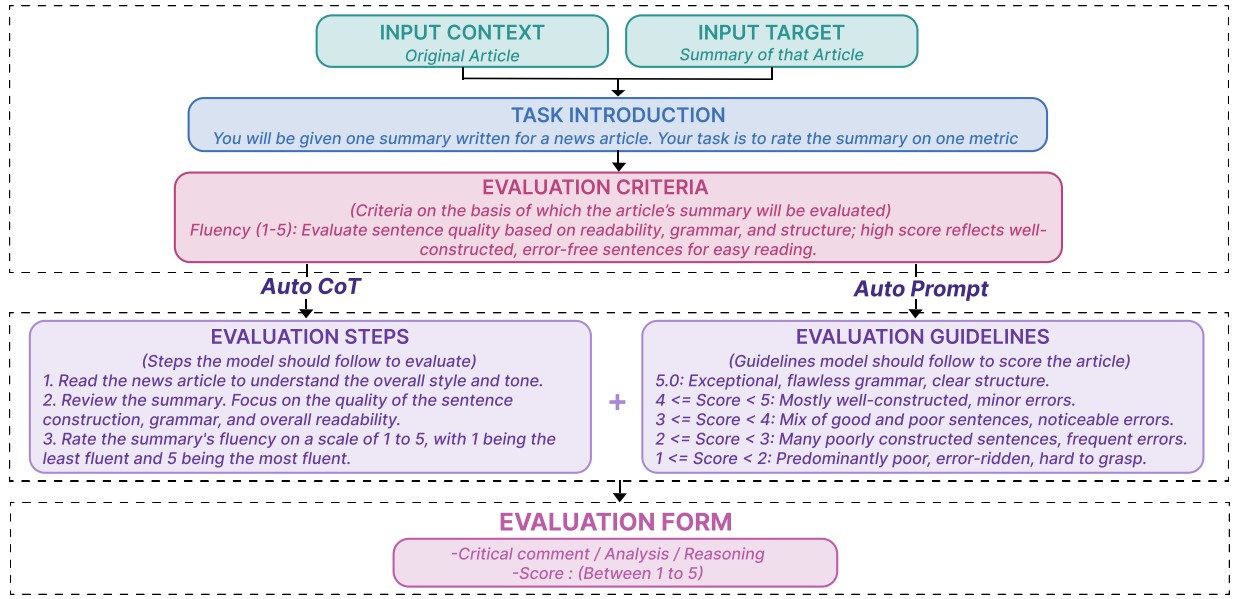

Figure 4: Prompting Schema

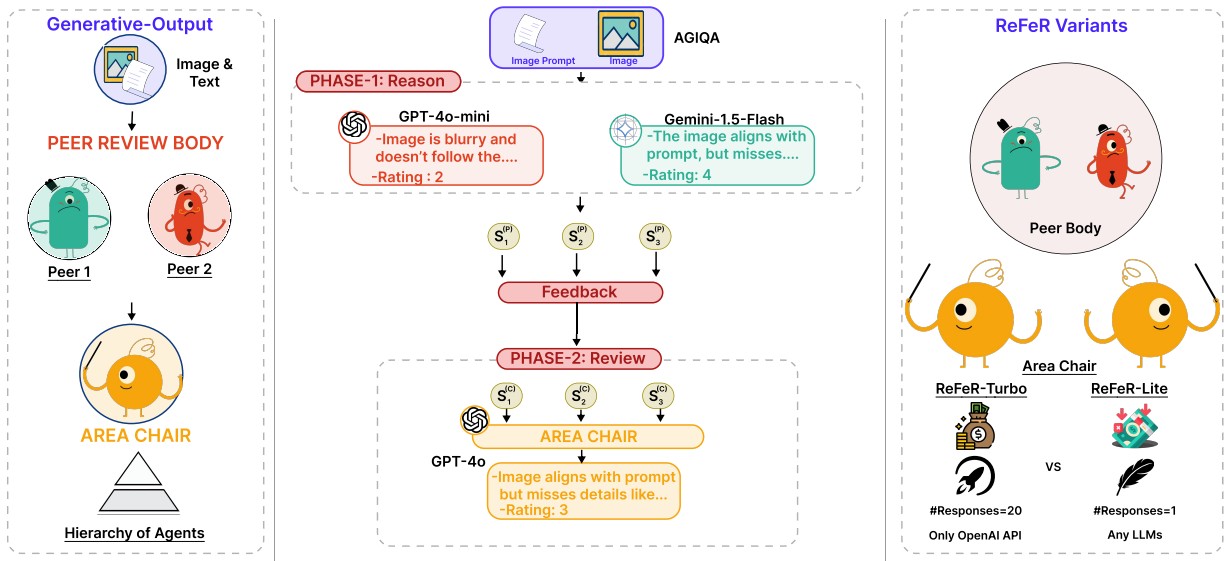

Figure 5: **Illustration of ReFeR for Multimodal evaluation shown on AGIQA dataset**. A similar version of ReFeR working on textual TopicalChat dataset is shown in Fig. 1.

# B ReFeR Algorithm

---

**Algorithm 1:** ReFeR Framework for Evaluating Generative Outputs

---

**Input :**

- Generative output $O$ (text or image)
- Prompting schema $E_P$ for peers and $E_{AC}$ for area chair
- Peer Models $P = \{P_1, P_2, P_3, \ldots, P_K\}$ (K Peers)
- Area Chair Model $AC$
- Variant $V \in \{\text{ReFeR-Pro}, \text{ReFeR-Lite}\}$
- Number of responses $n$ (only for ReFeR-Pro)

**Output:**

- Final Evaluation Score $S_{\text{final}}$

- Constructive Feedback $C_{\text{final}}$

**Phase 1: Peer Review Body Evaluation**;
**foreach** *peer model $P_i \in P$* **do**
    // Each peer independently evaluates $O$ using prompting schema $E_P$
    $(C_i, S_i) \leftarrow \text{Evaluate}_{P_i}(G, E_P)$;
**end**

**Phase 2: Area Chair Evaluation**;
**if** $V = \textit{ReFeR-Lite}$ **then**
    $n \leftarrow 1$;
**end**
**else**
    $n \leftarrow 20$;
**end**
**for** $j = 1$ **to** $n$ **do**
    $(C_{\text{AC}}^{(j)}, S_{\text{AC}}^{(j)}) \leftarrow \text{Evaluate}_{AC}^{(j)}(G, E_{AC}, \{(C_i, S_i)\}_{i=1}^{K})$;
**end**
// Compute final score
$S_{\text{final}} \leftarrow \frac{1}{n} \sum_{j=1}^{n} S_{\text{AC}}^{(j)}$;
$C_{\text{final}} \leftarrow$ Choose 1 randomly whose score is closest to $S_{\text{final}}(C_{\text{AC}}^{(1)}, \ldots, C_{\text{AC}}^{(n)})$;

**return** *Final Evaluation Score $S_{final}$, Constructive Feedback $C_{final}$*;

---

**Mathematical Notation Summary:**

- $G$: Generative output to be evaluated.
- $E_P$: Prompt of peer.
- $E_{AC}$: Prompt of area chair.
- $P_i$: Peer agent $i$, for $i = 1, \ldots, K$.
- $K$: Number of peer agents.
- $C_i$: Comment from peer agent $P_i$.
- $S_i$: Score from peer agent $P_i$.

- $AC$: area chair agent.
- $n$: Number of independent evaluations by $AC$ in ReFeR-Pro.
- $C_{\text{AC}}^{(j)}, S_{\text{AC}}^{(j)}$: Comment and score from the $j$-th evaluation by $AC$.
- $C_{\text{final}}, S_{\text{final}}$: Final comment and score.

In summary, the ReFeR framework formalizes the evaluation of generative outputs by modeling the process after the hierarchical peer review system, with mathematical rigor to facilitate clarity and reproducibility. This approach not only enhances the evaluation accuracy but also provides constructive feedback, aligning closely with human judgment and expectations in complex evaluation scenarios.

## C    Hyperparameters

For the ReFeR NLG Evaluation setup, following Analyze-Rate (Chiang & Lee, 2023), we set these hyper-parameters as follows, for the AreaChair GPT-4o-mini model- *temperature=1, max_tokens=256, top_p=1, frequency_penalty=0, presence_penalty=0, stop=None, n=20* (varies for ReFeR Lite and Pro). For the peer models, we use the default hyperparameters except for the *max_tokens=128*. For multimodal evaluation, we use the same setup for the AC, but for the peers, we increase the *max_tokens* from 128 to 192 tokens. For reasoning tasks, we follow the NLG evaluation setup for the area chair, but we don't set any limit on the *max_tokens* hyperparameter. For the peer models, we increase *max_tokens* to *256* and set the hyper-parameters *temperature=1, top_p=1*.

## D    Prompt Optimization

Prompt optimization methods utilizing LLMs, such as OPRO (Yang et al., 2024), APE (Zhou et al., 2023), ProTeGi (Pryzant et al., 2023), and SCULPT Kumar et al. (2024), often employ text-gradient or feedback-based techniques to refine prompts. These methods typically involve providing a capable LLM with error examples and obtaining feedback, which serves as a text gradient to adjust the prompt for improved performance. Such approaches have demonstrated effectiveness for short questions/requests and relatively simple tasks.

Table 8 compares our prompt with an optimized prompt using ProTeGi on the TopicalChat dataset. We utilized the default ProTeGi settings with GPT-4o-mini as the optimizer LLM. Our findings indicate that prompt optimization is time-consuming and incurs higher costs than evaluation itself due to the iterative improvement process of prompt optimization over a test set, and that too with long inputs for complex tasks like this. Moreover, our prompts with the proposed structure yield better correlation than the optimized prompts from ProTeGi.

We attribute this outcome to the limitations of these methods when dealing with extensive inputs, such as conversation history and responses in TopicalChat, where entire dialogues are provided to evaluate and rate NLG output on various metrics. Unlike G-Eval, which only provides scores, methods like Analyze-Rate and ours improve scores based on generated analyses. In these cases, both the analyses and ratings are crucial for understanding errors, as the singular numerical rating value offers insufficient insight into prompt issues. Additionally, even when detailed analyses are provided in multiple error examples for the prompt optimization, the gradient-based approach may struggle with long contexts, making it challenging for the model to identify specific prompt deficiencies and provide useful feedback.

Table 8: **Analysis of Prompt Optimization.** Comparison of Average (across 4 metrics) results for different prompts on TopicalChat dataset.

| Method | ProTeGi | | Ours | |
|---|---|---|---|---|
| | $\rho$ | $\tau$ | $\rho$ | $\tau$ |
| Llama-3.1-8B | 0.347 | 0.303 | 0.386 | 0.337 |
| Mistral-Nemo-12B | 0.387 | 0.336 | 0.464 | 0.402 |
| Gemma-2-9B | 0.511 | 0.444 | 0.563 | 0.489 |
| ReFeR Pro | 0.625 | 0.511 | 0.628 | 0.513 |

The table shows the results of average results across 4 metrics for peers and the framework for Prompt optimized by ProTeGi vs Our Prompt generated through our prompting schema. We can see that the peers' performance declines with the optimized prompts. These prompts were the best prompts after 3 rounds of Prompt-Optimization with ProTeGi. But still their performance falls short to our prompting schema. And even though the framework is relatively close, it would still make a point on how the effort and costs for the prompt optimizations would not be worth it. Running ProTeGi prompt optimizations alone for peers and area chair costs $\sim$ 4 times the cost of evaluating using ReFeR-Pro.

### Prompt Ablation

Table 9 shows the ReFeR framework performance on TopicalChat Dataset with different prompts. We can see from the results that our prompting Schema gives the highest performance with Analyze-rate being the second. From the significant difference, we can see that G-eval scores only prompting doesn't work for our ReFeR framework.

Table 9: **Analysis of Prompt Ablation.** Average results (across 4 metrics) on TopicalChat Dataset of ReFeR-Pro with different Prompt structure.

| | Prompt Schema | Average | |
|---|---|---|---|
| | | $\rho$ | $\tau$ |
| **ReFeR-Pro** | G-Eval | 0.568 | 0.454 |
| | Analyze-Rate | 0.592 | 0.510 |
| | Ours | 0.628 | 0.513 |

## E  Auto Prompt

**Example Implementation of Autoprompt:**

**Input AutoPrompt:**

---

*AutoPrompt for Engagingness Evaluation for TopicalChat*

*You are tasked with creating a clear and concise prompt for a task based on the provided prompt structure and examples from dataset. The prompt should be written in such a way that it can be easily understood and followed by another LLM or human user performing the task. Your prompt should include the following:*

- *A brief overview of the task.*
- *Evaluation criteria explaining what metric the evaluation is going to be on.*
- *Clear instructions for how to approach the task or evaluation steps.*
- *Use the examples of dataset, analyze and understand how it is evaluated for the given metric and give a detailed Evaluation guidelines which will tell when to give a particular score.*

*Example Prompt Structure:*
*You will be presented with a conversation between two individuals and given a potential response for the next turn in the conversation, along with a fact that the response is based on.*
*Your task is to evaluate the response on a single metric: Engagingness. The rating must be given after giving the analysis too.*
*Evaluation Criteria:*
*Engagingness (1-3): Assess whether the response is dull, moderately interesting, or highly engaging.*
*Evaluation Steps:*

   *1. Read the conversation, the corresponding fact and the response carefully.*

   *2. Rate the response on a scale of 1-3 for engagingness, according to the criteria above.*

*Please ensure the prompt explains the rating scale from 1 to 3 clearly.*

*Examples for Task:*
*Example 1:*

*Conversation History:*
*Person A: "I heard that flamingos get their pink color from their diet."*
*Person B: "Really? I didn't know that! What do they eat to turn pink?"*
*Fact: Flamingos get their pink color from consuming shrimp and algae rich in carotenoids.*
*Response: "Yes! They eat shrimp and algae, which contain pigments called carotenoids that give them their pink color."*
*Rating: 3*

*Example 2:*

---

*Conversation History:*

*Person A: "I just rewatched Interstellar. The way they depict time dilation near a black hole is incredible."*

*Person B: "Yeah, the science behind it is pretty solid. What did you think of the emotional storyline with Cooper and his daughter?"*

*Person A: "I thought it was powerful how they tied in the human element with such a complex scientific concept."*

*Person B: "Do you think the film's portrayal of black holes was accurate?"*

*Fact: Interstellar consulted physicist Kip Thorne to ensure the black hole depiction was as scientifically accurate as possible, especially with the time dilation effects.*

*Response: "It was fine."*

*Rating: 1*

The above prompt is an example of Auto Prompt for the Engagingness metric in TopicalChat dataset. The text marked in blue, i.e., the example prompt structure and the examples are the inputs to be given by the user to get the evaluation guidelines to be used in our proposed prompting schema. In the example above we use the prompt structure from Analyze-Rate engagingness prompt. Hence by prompting a capable LLM with an Auto Prompt crafted for a task giving the prompt structure and examples, we can obtain a prompt following our prompting schema for that task, including evaluation guidelines.

**Output based on the above prompt:**

---

***Engagingness Peer Prompt***

*You will be given a conversation between two individuals. You will then be given one potential response for the next turn in the conversation. The response concerns an interesting fact, which will be provided as well.*

*Your task is to rate the responses on one metric. Please make sure you read and understand these instructions carefully. Please keep this document open while reviewing, and refer to it as needed.*

*Evaluation Criteria: Engagingness (1-3): Assess whether the response is dull, moderately interesting, or highly engaging.*

*Evaluation Steps:*

    *1. Read the conversation, the corresponding fact and the response carefully.*

    *2. Rate the response on a scale of 1-3 for engagingness, according to the criteria above.*

*Evaluation Guidelines:*
*- A score of 1 (dull) means that the response is generic and dull.*
*- A score of 2 (somewhat interesting) means the response is somewhat interesting and could engage you in the conversation (e.g., an opinion, thought).*
*- A score of 3 (interesting) means the response is very interesting or presents an interesting fact.*

---

## F  SummEval Results

We test our framework on the SummEval Dataset, comparing it with the baselines G-Eval and Analyze-Rate. We first show our individual peer performances, then the baselines, and finally, the two variants of our ReFeR framework. Before delving deep into the results of this benchmark, it is important to discuss the dataset distribution of SummEval. As shown in Fig. 6, the dataset is highly skewed for the consistency and fluency metrics, with almost 1300+ and 1100+ samples having a score of 5 for consistency and fluency, respectively.

The skewed distribution in the Summeval dataset creates an imbalance in correlation evaluations. Smaller models, such as Gemma-2-9B, which may lack sensitivity to subtle differences in the data (e.g., article and

Table 10: **Performance analysis on SummEval dataset.** Comparison of various methods for NLG evaluation on SummEval.

| | Method | Coherence | | Consistency | | Fluency | | Relevance | | Average | |
|---|---|---|---|---|---|---|---|---|---|---|---|
| | | $\rho$ | $\tau$ | $\rho$ | $\tau$ | $\rho$ | $\tau$ | $\rho$ | $\tau$ | $\rho$ | $\tau$ |
| Peer Agents | Llama-3.1-8B | 0.351 | 0.287 | 0.425 | 0.381 | 0.307 | 0.277 | 0.361 | 0.295 | 0.361 | 0.310 |
| | Mistral Nemo-12B | 0.367 | 0.296 | 0.383 | 0.340 | 0.239 | 0.211 | 0.368 | 0.303 | 0.339 | 0.287 |
| | Gemma-2-9B | **0.560** | **0.460** | 0.474 | **0.433** | 0.387 | **0.347** | 0.517 | 0.422 | 0.484 | **0.415** |
| Baselines | Analyze-Rate | 0.533 | 0.392 | 0.382 | 0.305 | 0.353 | 0.283 | 0.430 | 0.320 | 0.425 | 0.325 |
| | G-Eval | 0.509 | 0.387 | 0.475 | 0.386 | 0.334 | 0.290 | **0.571** | **0.433** | 0.472 | 0.374 |
| Ours | ReFeR | 0.528 | 0.403 | **0.478** | 0.390 | **0.425** | 0.342 | 0.521 | 0.395 | **0.488** | 0.382 |
| | ReFeR Lite | 0.483 | 0.400 | 0.472 | 0.420 | 0.360 | 0.324 | 0.472 | 0.397 | 0.447 | 0.385 |

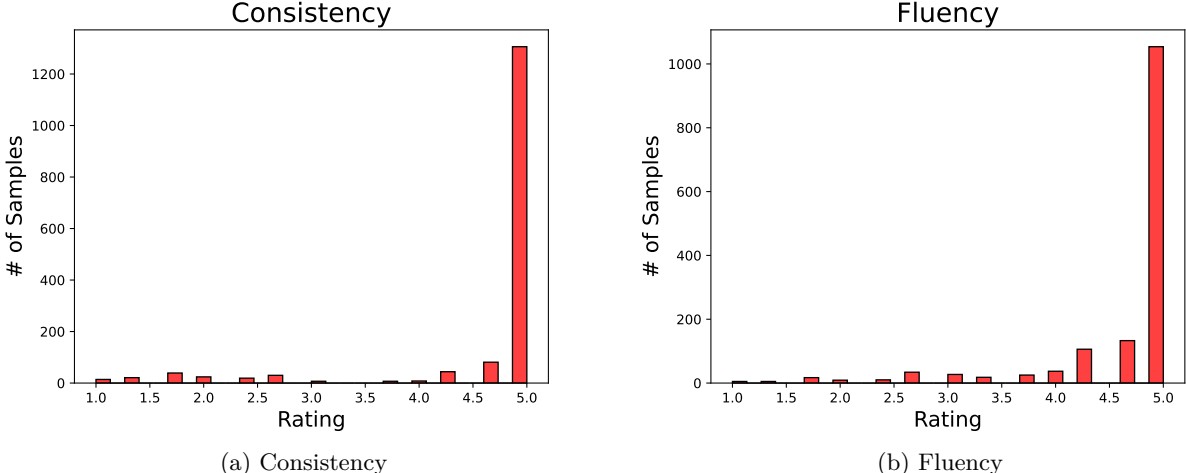

(a) Consistency         (b) Fluency

Figure 6: **Ratings Distribution.** We show the distribution of human annotations for Consistency, FLuency metrics in the SummEval dataset

summary), often give uniformly high scores that mimic the skewed human annotations, resulting in higher correlations. However, this does not reflect the model's true ability to understand and follow instructions. In contrast, larger models like GPT-4o-mini, which adhere more strictly to evaluation guidelines, tend to generate more varied scores. This often leads to lower correlations due to the skewed nature of the human annotations. Additionally, in cases where the ReFeR framework provides consistent scores, the result is a high number of tied predictions, which further lowers Kendall's tau coefficient due to the large number of tied pairs. This complicates the interpretation of performance for larger models and more advanced frameworks, as the lack of variability in the dataset hampers an accurate assessment of model effectiveness. Given these challenges, although SummEval is a popular benchmark dataset used for NLG evaluation, we do not consider Summeval to be an appropriate benchmark for testing our methods, unless a uniformly distributed sample can be extracted—a difficult task given the inherent skewness of the annotations. Hence, even though ReFeR-Pro outperforms other baselines on this dataset, we present these results only in the appendix.

# G Datasets

**NLG Evaluation**

- **SummEval** (Fabbri et al., 2021) provides human assessments on four critical dimensions of summarization quality: Coherence, Consistency, Fluency and Relevance, utilizing the CNN/DailyMail dataset (Hermann et al., 2015) as its foundation.
- **TopicalChat** (Gopalakrishnan et al., 2019) is a dataset of conversations. We use the dataset created by Mehri & Eskenazi (2020) using the TopicalChat dataset in which they give a possible next response generated by a language model for a given conversation history, and the human annotation score of the

response on five attributes: Coherence, Engagingness, Groundedness, Naturalness, and Understandability. We exclude Understandibility, following the previous works G-Eval, Analyze-Rate and Uni-Eval[4]

**MultiModal Evaluation**

- **ICQD** (Image Caption Quality Dataset) (Levinboim et al., 2019) focuses on the task of Quality Estimation (QE) for image captions. We use the test dataset which provides human ratings (0/1) on quality. We scale these average ratings to a scale of 0-100 for our evaluation.
- **AGIQA** (AI Generated Image Quality Assessment) (Zhang et al., 2023) presents a AGI quality assessment database, AGIQA-1K, which consists of 1,080 AGIs generated from diffusion models. They provide MOS (Mean Opinion Score) in the range of 0-5. We have observed that the dataset is skewed around certain scores around 3-3.5. So to test on a subset which has variance of image quality ratings, we select 500 samples, such that the data more or less equally spread on the rating range (0-5).

**Reasoning**

- **AQuA** (Algebra Question Answering) (Ling et al., 2017) dataset is designed to assess a model's reasoning abilities in solving algebraic word problems. It consists of multiple-choice math questions, where the model must understand and compute the correct answer from several options.
- **BBH-DU** (Big Bench Hard Date Understanding) (Srivastava et al., 2023) dataset is part of the BIG-Bench benchmark. It focuses on testing a model's ability to comprehend and reason about date-related information, such as calculating durations and interpreting dates.
- **CSQA** (CommonsenseQA) (Aggarwal et al., 2021) dataset is designed to test a model's understanding of commonsense knowledge through multiple-choice questions. Each question requires reasoning over general world knowledge, with answer choices based on various plausible but nuanced options, testing the model's ability to pick the most commonsensical answer.
- **GSM8k** (Grade School Math 8K) (Cobbe et al., 2021) dataset is a collection of 8,000 challenging grade-school-level math word problems. It is designed to test a model's ability to perform multi-step arithmetic reasoning and solve math problems requiring logical thinking and numerical computation.

## H  Note on ReConcile

ReConcile (Chen et al., 2024a) is another relevant multi-agent framework that utilizes different LLMs with similar capabilities to engage in discussions and reach consensus. However, we exclude ReConcile from our baselines because its use of 3 LLMs of similar capabilities and makes it an unfair comparison to our framework, which employs 3 smaller models as peers and 1 larger model as the area chair. Simulating ReConcile with our setup would require excluding one of the models, either from the peer group or the area chair, which would lead to an unbalanced debate. In particular, if we use 2 smaller models and a large model, the debate would be dominated by the larger model, resulting in biased outcomes. For these reasons, we do not include ReConcile as a direct baseline.

## I  More Related Works

In this section we discuss more recent works in the field of multi-agent reasoning. While our ReFeR framework proposes a specific two-level hierarchy inspired by academic peer review, the broader field of multi-agent reasoning has explored several alternative collaborative architectures. Many recent works focus on achieving consensus through unstructured or semi-structured debate. For example, ReConcile (Chen et al., 2024a) implements a "round-table conference" where diverse LLM agents iteratively discuss and refine their answers to reach a consensus, contrasting with ReFeR's single-pass hierarchical synthesis. Similarly, CONSENSAGENT (Pitre et al., 2025) focuses on mitigating "sycophancy" in multi-agent debate through dynamic prompt refinement to achieve more efficient and effective consensus, whereas ReFeR's structure bypasses the need for peer consensus by vesting final authority in the area chair. Other frameworks introduce adaptive mechanisms to balance performance and efficiency. DOWN (Debate Only When Necessary) (Eo et al., 2025) proposes a dynamic framework that triggers a multi-agent debate only when an initial agent's confidence score is low, offering a different approach to efficiency than ReFeR-Lite's fixed, lightweight structure. Another direction

---

[4]Uni-Eval shows results on the 4 metrics and uses the Understandability metric for transfer experiment, hence only 4 dimensons are shown in the following works. Refer to Zhong et al. (2022) for more details.

involves diverse reasoning paths and role-playing. CoMM (Chen et al., 2024b) prompts agents to adopt different expert roles, encouraging varied problem-solving strategies, which differs from ReFeR's functionally defined roles of peer evaluator and synthesizer. Lastly, some frameworks integrate external knowledge, such as MADKE (Wang et al., 2024a), which provides agents with a shared knowledge pool to enhance their debating capabilities—a clear distinction from ReFeR's reliance on the internal knowledge of its constituent models. These kind of reasoning frameworks can be used for specific use cases where you have a defined knowledge pool.

## J Finetuning

Utilizing Analysis from larger LLMs ("Area Chair"), we enhance smaller LLMs through instruction-tuning, using a dataset crafted from comprehensive evaluations. We use the analysis feedback generated within the ReFeR framework, transforming it into a useful resource for instructional tuning. This fine-tuning significantly improves smaller models performance, enabling them to reach or surpass their larger counterparts in evaluation tasks. We use Mistral-7B, since it can be easily deployable on a small GPU and finetune. We used the instruction-tuning dataset (final output of Area Chair) of ReFeR framework as the training data.

Table 11: **Improving smaller models via instruction-tuning.** Finetuning Results for Mistral-7B model on TopicalChat Dataset

| Model | Coherence | | Engagingness | | Groundedness | | Naturalness | | Avg | |
|---|---|---|---|---|---|---|---|---|---|---|
| | $\rho$ | $\tau$ | $\rho$ | $\tau$ | $\rho$ | $\tau$ | $\rho$ | $\tau$ | $\rho$ | $\tau$ |
| Mistral-7B No Finetune | 0.124 | 0.102 | 0.167 | 0.134 | 0.078 | 0.069 | 0.100 | 0.081 | 0.117 | 0.096 |
| Mistral-7B Finetuned | 0.457 | 0.348 | 0.626 | 0.486 | 0.487 | 0.437 | 0.493 | 0.377 | 0.516 | 0.412 |

## K Communication Strategies:

Table 12: **Communication Strategies.** Results on TopicalChat showing different generation and communication strategies for ReFeR-Pro.

| Communication | Coherence | | Engagingness | | Groundedness | | Naturalness | | Avg | |
|---|---|---|---|---|---|---|---|---|---|---|
| Peer Feedback to AreaChair | $\rho$ | $\tau$ | $\rho$ | $\tau$ | $\rho$ | $\tau$ | $\rho$ | $\tau$ | $\rho$ | $\tau$ |
| Comment Only | **0.602** | **0.471** | 0.635 | 0.502 | **0.661** | **0.590** | 0.587 | 0.454 | 0.621 | 0.504 |
| Score Only | 0.585 | 0.454 | **0.673** | **0.535** | 0.628 | 0.577 | **0.625** | **0.484** | **0.628** | **0.513** |
| Both Comment & Score | 0.580 | 0.453 | 0.642 | 0.512 | 0.605 | 0.545 | 0.555 | 0.427 | 0.596 | 0.484 |

The type of feedback provided by peers to the area chair plays a crucial role in determining overall effectiveness. We explored three communication strategies: passing only scores, passing only comments, and passing both comments and scores. Table 12 presents the impact of different feedback strategies on the framework's performance. The results indicate that passing only scores to the AC yields the best performance, with passing only comments being a close second. This is likely because when both comments and scores are passed, the AC model becomes more prone to confusion due to conflicting analyses or scores, and the longer prompt inputs negatively affect its decision-making Liu et al. (2023b). Based on these findings, we adopt the scores-only strategy for all subsequent experiments with our framework.

## L Inference and Computation

Fig. 7 presents the time taken per instance for ReFeR Variants and baseline models. G-Eval demonstrates the fastest inference speed, as it only generates scores. In contrast, Analyze-Rate takes nearly twice as long as G-Eval, since it produces both an analysis and a rating. ReFeR-Lite and ReFeR-Pro require only approximately 1.5 times the duration of Analyze-Rate, despite being multi-model frameworks. Notably, there is minimal difference between the Lite and Pro variants due to the influence of the $n$ hyperparameter, indicating that the bulk of the processing time arises from the involvement of multiple models in the framework.

Although we show the monetary cost comparison of our framework vs the other methods, it is not the only cost for evaluation. We need to also consider test-time compute metric like FLOPs processed. So as to know all parameter counts and calculate the FLOPs processed for each method, we use an open source model

Figure 7: **Inference speed comparison with baselines**

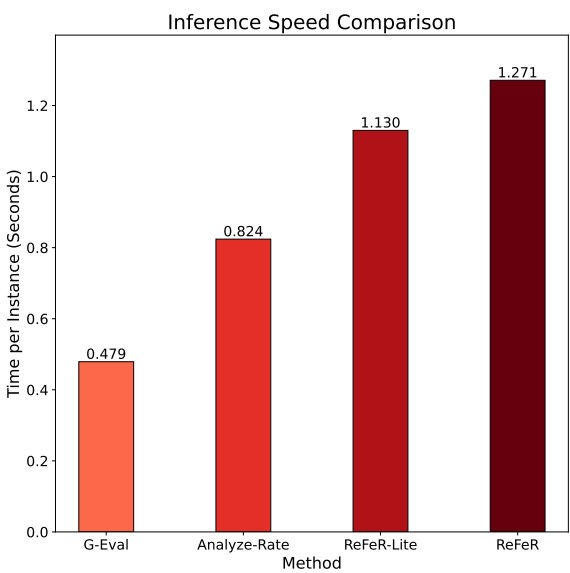

Qwen-2.5-72B, as our AC model (with the same peer models) and for G-Eval & Analyze-rate. We calculate the FLOPs with this approximation formula:

$$\text{FLOPs} = \text{layers} \times \left[ 4 \cdot \text{sequence\_length} \cdot d_{\text{model}}^2 + 8 \cdot d_{\text{model}} \cdot \text{ffn\_dim} \right]$$

The components involved in the FLOPs calculation are as follows: **layers** refers to the number of transformer layers, **sequence_length** is the length of the input sequence, **d_model** is the dimensionality of the model (hidden size), and **ffn_dim** represents the dimensionality of the feed-forward network, which is typically 4 times $d_{\text{model}}$.

For the self-attention mechanism, the cost of generating the query, key, and value matrices is given by:

$$3 \times d_{\text{model}}^2 \times \text{sequence\_length}.$$

The cost of the scaled dot-product attention, which involves computing attention scores and applying them, is approximately:

$$d_{\text{model}}^2 \times \text{sequence\_length}.$$

Therefore, the total cost for the self-attention mechanism is:

$$4 \times d_{\text{model}}^2 \times \text{sequence\_length}.$$

In the feed-forward network (FFN), which consists of two linear layers with a ReLU activation in between, the cost for each linear layer is approximately:

$$2 \times d_{\text{model}} \times \text{ffn\_dim}.$$

Thus, the total cost for the FFN, considering both the forward and backward passes, is:

$$8 \times d_{\text{model}} \times \text{ffn\_dim}.$$

Although the exact model architectures vary for different models, and hence the actual FLOPs vary, but we believe this approximation formula helps to give enough information for comparison of the computation. The given Total Input tokens is for all 4 dimensions(coherence,engagingness,etc) for each model/method. As mentioned, we use the same model Qwen-2.5 as our AC and for G-Eval, Analyze-Rate.

We modulate the performance with the hyperparameter *'n'* to see whether our method consistently is above the baselines in the performance-compute trade off and we present these results in Table 14. But since the open models don't have 'n' hyperparamater, we do $n$ calls to get the $n$ responses.

Table 13: Model Comparison of FLOPs. ReFeR-Lite is calculated as the cost of the peer models(first 3 rows) and AC. ReFeR-Pro is calculated as summation of the peers and 20 AC calls. G-Eval, Analyze-Rate are calculated as 20 calls.

| Model | $d_{\mathbf{model}}$ | ffn__dim | layers | Total Input tokens | FLOPs ($\times 10^{15}$) |
|---|---|---|---|---|---|
| Llama-3.1-8B | 4096 | 14336 | 32 | 970720 | 2.11 |
| Mistral-Nemo-12B | 5120 | 14336 | 40 | 970720 | 4.11 |
| Gemma-2-9B | 3584 | 28672 | 42 | 970720 | 2.14 |
| AC (Qwen-2.5-72B) (n=1) | 8192 | 29568 | 80 | 1016279 | 22.05 |
| Analyze-Rate(n=20) | 8192 | 29568 | 80 | 856100 | 372.16 |
| G-Eval(n=20) | 8192 | 29568 | 80 | 888500 | 386.07 |
| ReFeR-Pro | 8192 | 29568 | 80 | - | 449.31 |
| ReFeR-Lite | 8192 | 29568 | 80 | - | 30.40 |

Table 14: Performance and Computation of methods across different $n$-values. $\rho$ is average spearman correlation across all metrics.

| n | Method | $\rho$ | FLOPs ($\times 10^{15}$) |
|---|---|---|---|
| 1 | ReFeR-Lite | 0.620 | 30.41 |
| | Analyze Rate | 0.545 | 18.61 |
| | G-Eval | 0.608 | 19.3 |
| 3 | ReFeR | 0.639 | 74.51 |
| | Analyze Rate | 0.547 | 55.83 |
| | G-Eval | 0.626 | 57.9 |
| 5 | ReFeR | 0.646 | 118.61 |
| | Analyze Rate | 0.554 | 93.05 |
| | G-Eval | 0.633 | 96.5 |
| 8 | ReFeR | 0.648 | 184.76 |
| | Analyze Rate | 0.542 | 148.88 |
| | G-Eval | 0.636 | 154.4 |
| 10 | ReFeR | 0.649 | 230.5 |
| | Analyze Rate | 0.541 | 186.1 |
| | G-Eval | 0.637 | 193.0 |

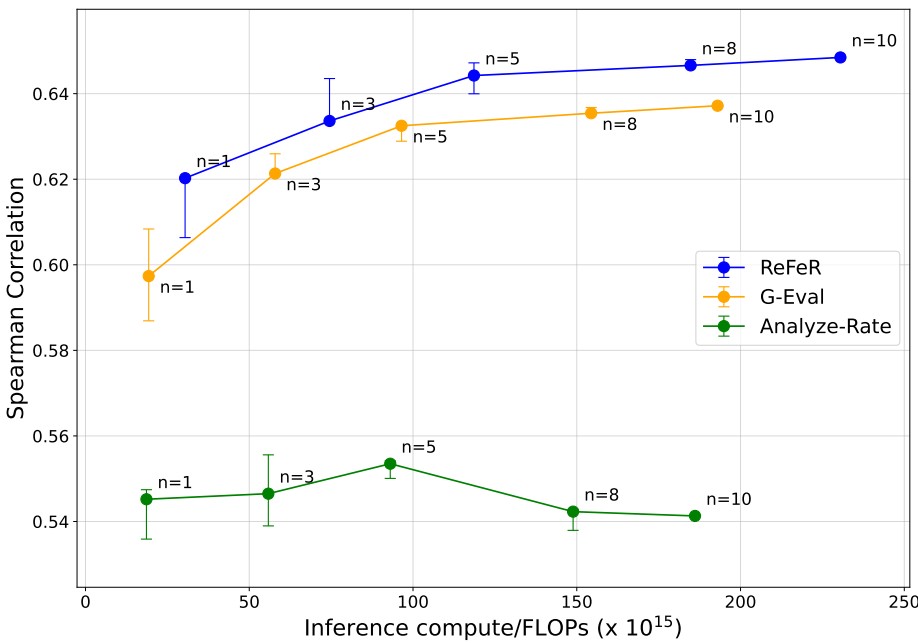

Figure 8: **Performance vs Compute Cost**

For better understanding and visualization, we show the performance (spearman correlation) vs FLOPs in Figure 8.

From Figure 8 and Table 14, we can observe that our ReFeR method is above the other methods in terms of performance-compute trade off. We can see that although ReFeR is relatively computationally costlier than the other methods, it has achieved higher correlations with relatively lower values of n, as we see that n=8 of ReFeR gives better performance than n=10 of G-eval while maintaining lesser FLOPs compared to G-Eval's n=10. We observe overall that ReFeR-Pro (or n=10 here,) has more FLOPS than G-Eval, but ReFeR gives more correlation than G-Eval. Whereas ReFeR-Lite (n=1) has significantly lesser FLOPs but gave very high correlation although didn't cross in this experiment with Qwen-2.5-72B model. But, we can also see some other observations here in this setting with an open source model. Both ReFeR and G-Eval follow the same trend where we see an increase in performance with increase in 'n', but we don't see this trend in Analyze-rate. This leads us to the observation that some prompting schema are not consistent across different models.

Overall we do think that ReFeR-Lite is significantly better than other methods considering the overall FLOPs to performance ratio.

## M   Statistical Tests

Table 15: **Statistical Tests.** p-values for statistical tests between ReFeR Pro and baselines on TopicalChat.

| Metric | ReFeR vs. Analyze rate | ReFeR vs. G-Eval |
|---|---|---|
| Coherence | $2.34 \times 10^{-6}$ | $4.29 \times 10^{-13}$ |
| Engagingness | $3.70 \times 10^{-5}$ | $1.01 \times 10^{-17}$ |
| Groundedness | $1.15 \times 10^{-7}$ | $6.31 \times 10^{-12}$ |
| Naturalness | $1.19 \times 10^{-4}$ | $0.0736$ |

We conducted the paired t-test to compare our method with the baseline and we report the p-values. As we can see for only one metric in G-Eval, the p-value is greater than the typical significance level (0.05). Hence, from the p-values, we can see that ReFeR is, in general, statistically significantly better than G-Eval and Analyze Rate.

## N   Example of Reasoning improved by Framework

---

### *Example from GSM8K*

**Question:** *Solo has to read 4 pages from his Science textbook, 20 pages from his Social Studies textbook, 7 pages from his History textbook and 8 pages from his Geography textbook. Solo read 15 pages on Monday. If he has 4 more days to complete all his reading, how many pages does he need to read, on average, in one day?*

**Gold Answer:** *6 pages*

**Peer 1's Response:** *To find the total number of pages Solo needs to read, we add up the pages from each textbook: 4 (Science) + 20 (Social Studies) + 7 (History) + 8 (Geography) = 39 pages. Solo needs to read, on average, 9.75 pages per day.* ***Answer: 9.75***

**Peer 2's Response:** *Total pages to read: 4 (Science) + 20 (Social Studies) + 7 (History) + 8 (Geography) = 40 pages. After subtracting 15 pages already read, Solo needs to read 25 more pages. He needs to read an average of 6.25 pages per day.* ***Answer: 6.25***

**Peer 3's Response:** *Total pages to read: 4 (Science) + 20 (Social Studies) + 7 (History) + 8 (Geography) = 39 pages. Solo needs to read, on average, 9.75 pages per day.* ***Answer: 9.75***

**Area Chair's Response:** *Total pages to read: 4 + 20 + 7 + 8 = 39 pages. After subtracting 15 pages already read, Solo needs to read 24 more pages. He needs to read an average of **6 pages** per day. LLM 1 and LLM 3 provided the correct calculations but made a mistake in their final answers by not subtracting the number of pages Solo read on Monday from the total number of pages. They both stated that Solo needs to read, on average, 9.75 pages per day, which is incorrect. LLM 2 made a calculation mistake. My solution corrects these errors and provides the accurate calculation for the average number of pages Solo needs to read per day.* **Answer: 6**

---

---

**Example 2 from GSM8K**

**Question:** *Conor lives near a beach and loves going there every day to have fun. On a particular week, he found 50 people at the beach on the first day. The next day, 20 more people were present at the beach than on the first day, and on the third day, the total number of people increased to twice the number that were there on the second day. If Conor saw an average of 60 people each day the rest of the week, calculate the total number of people Conor saw at the beach that week.*

**Gold Answer:** *500*

**Peer 1's Response:**

*Analysis: To solve this problem, I will first determine the number of people present at the beach on each of the first three days. Then, I will calculate the total number of people Conor saw at the beach for the entire week by adding the number of people on each of the first three days to the average number of people he saw each day for the rest of the week. Since there are 5 more days in a standard week, I will multiply the average daily number of people by 5 to find the total number of people Conor saw for the remaining days. Finally, I will add the total number of people seen on the first three days to the total number of people seen during the rest of the week to find the overall total.*
*The first three days:*
*- Day 1: 50 people*
*- Day 2: 20 more people than on the first day, so 50 + 20 = 70 people*
*- Day 3: Twice the number of people on the second day, so 2 x 70 = 140 people*
*Total number of people on the first three days: 50 + 70 + 140 = 260*
*Average number of people on the rest of the week (5 days): 60 people/day*
**Answer: 560**

**Peer 2's Response:**

*Analysis: On the first day, there were 50 people at the beach. On the second day, there were 20 more people than the first day, making it 70 people. On the third day, the number of people doubled from the second day, resulting in 140 people. For the remaining 4 days, Conor saw an average of 60 people each day, so the total for these days is 4 * 60 = 240 people. Adding all these together gives the total number of people Conor saw that week.* **Answer: 450**

**Peer 3's Response:**

*Analysis: First, we calculate the number of people at the beach on the second and third days. The second day had 50 + 20 = 70 people. The third day had 70 * 2 = 140 people. Then we calculate the total number of people on the remaining five days as 60 * 5 = 300. Finally, we add up the people from all seven days: 50 + 70 + 140 + 300 = 560.*
**Answer: 560**

**Area Chair's Response:**
*Analysis: Let's break down the problem step by step.*

1. *On the first day, Conor saw 50 people at the beach.*

2. *On the second day, there were 20 more people than the first day, so the total for the second day is: 50 + 20 = 70 people.*

3. *On the third day, the number of people doubled compared to the second day, giving us: $70 \times 2 = 140$ people.*

---

> *4. For the remaining four days of the week, Conor saw an average of 60 people each day. The total for these four days is:* $60 \times 4 = 240$ *people.*
>
> *Now, let's add all these numbers together to find the total number of people Conor saw at the beach over the week: 50 (day 1) + 70 (day 2) + 140 (day 3) + 240 (remaining days) = 500.*
> *Ample insights were provided by the LLMs, particularly the correct calculation for each day. However, LLM 2 had a calculation mistake when reporting the final total as 450,which is inaccurate. LLM 3 correctly adds up the totals but does not account for the final calculation correctly, as it appears to calculate five days instead of the four remaining after the first three. Thus, compiling our information correctly gives us the right total:* **Answer: 500**

## O   Hallucination

To understand the difference in response quality for our framework and the baselines, we used the HHEM-2.1 model by Bao et al. (2024). The HHEM-2.1 model measures the hallucination in the hypothesis (generated text) based on the premise (reference text). We did this analysis on the GSM8k experimental results, as the dataset provides the human annotated reasoning/rationale along with the answers. Hence we use this reference human reasoning as the premise and the generated reasoning in the final responses by different methods as the hypotheses. HHEM-2.1 gives scores in the range *(0,1)* where, 0 means that the hypothesis is not evidenced at all by the premise and 1 means the hypothesis is fully supported by the premise. Hence, the higher the HHEM score is, the lower the method's responses are hallucinated. We weren't able to test this on other benchmarks where human annotation reasoning/evaluation is not given.

Table 16: Comparison of average hallucination scores across methods on the GSM8k Benchmark. GPT-4o-mini is the HHEM score of GPT-4o-mini using our peer setup.

| Method | HHEM Score |
|---|---|
| GPT-4o-mini | 0.297 |
| Zero-Shot-CoT | 0.115 |
| Self Correction | 0.136 |
| Multi-Agent Debate | 0.102 |
| Multi-Agent Peer Review | 0.108 |
| ReFeR | **0.330** |

We can see that ReFeR clearly outperforms all the baselines and GPT-4o-mini's(with our peer setup) hhem score. Considering the GPT-4o-mini's score as baseline, we can see that ReFeR has reduced hallucination, whereas the other baseline methods have increased hallucination significantly. This shows that ReFeR produces better rationale in responses compared to the baselines.

## P   Homogeneous Experiments

We have demonstrated ReFeR primarily as a heterogeneous framework, where we leverage different models for peers and AC. Hence, we did this homogeneous experiment to test the framework's effectiveness when we use the same model for both the peers and AC. We use our best peer model Gemma2-9B as all peers & AC for this experiment, and to have diversity in responses among the peers and AC, we use the temperature hyperparameter, choosing temperatures *[0.25,0.5,0.75]* for the peer models. We retain the *temperature=1* for AC from our original setup. We can see the results for the homogeneous experiment in Table 17.

Table 17: Performance of ReFeR framework with same model, Gemma2-9B as both peers and AC. Varying temperatures were used for diversity among peers.

| Method | Coherence | | Engagingness | | Groundedness | | Naturalness | | Average | |
|---|---|---|---|---|---|---|---|---|---|---|
| | $\rho$ | $\tau$ | $\rho$ | $\tau$ | $\rho$ | $\tau$ | $\rho$ | $\tau$ | $\rho$ | $\tau$ |
| Gemma (*temp=0.25*) | 0.559 | 0.475 | 0.614 | 0.525 | 0.565 | 0.529 | 0.536 | 0.446 | 0.568 | 0.494 |
| Gemma (*temp=0.5*) | 0.548 | 0.465 | 0.611 | 0.525 | 0.571 | 0.535 | 0.540 | 0.452 | 0.568 | 0.494 |
| Gemma (*temp=0.75*) | 0.547 | 0.463 | 0.626 | 0.536 | 0.582 | 0.545 | 0.509 | 0.426 | 0.566 | 0.492 |
| ReFeR-Pro | **0.587** | 0.468 | **0.681** | 0.543 | **0.628** | **0.572** | **0.597** | 0.466 | **0.623** | 0.512 |
| ReFeR-Lite | 0.563 | **0.476** | 0.648 | **0.557** | 0.614 | 0.557 | 0.574 | **0.480** | 0.600 | **0.517** |

As we can see the framework has still shown to improve the performance using the same model for both peers and AC. Hence, we can see that the framework is effective even with using same model(varying temperatures) for peers and AC. This setting is particularly helpful when one wants to utilize a single model on the GPU for the entire evaluation.

## Q   Additional Experiments

**Averaging Ensemble**

We also considered an ensemble method where we combine (average) the higher performing AC model with the peers by giving higher weightage to the AC and see how this kind of direct numerical ensembling gives result, compared to our hierarchical review framework. In this ensemble method results in Table 18 where AC weightage is given, the remaining weightage (out of 1) will be split equally between the remaining peers, Equal weightage refers to the case where all the Peers and AC model are given equal weightage. As the weightages increase, the performance increases ever so slightly but plateaus later. We can see that this baseline still falls short to some of the other baselines like G-Eval & Analyze-Rate. This shows how simple ensembling like averaging of a higher models' evaluation might not work.

Table 18: Average Spearman and Kendall-Tau correlations across metrics for different average ensemble weightages on TopicalChat. $AC_w$ refers to the weightage of the AC in the average and the remaining weightage is equally split between the peers.

|  | Method | Spearman ($\rho$) | Kendall-Tau ($\tau$) |
|---|---|---|---|
| Peer Agents | Llama-3.1-8B | 0.386 | 0.337 |
|  | Mistral-Nemo | 0.464 | 0.402 |
|  | Gemma-2-9B | 0.563 | 0.489 |
| Baselines | G-Eval | 0.592 | **0.516** |
|  | Analyze-Rate | 0.591 | 0.514 |
| Area Chair | ReFeR-Lite | **0.602** | 0.503 |
|  | AC with Peer Setup | 0.5661 | 0.4932 |
|  | Peer Average | 0.539 | 0.447 |
| Average Ensemble | Equal Weightage | 0.5787 | 0.4735 |
|  | ($AC_w = 0.4$) | 0.5864 | 0.4781 |
|  | ($AC_w = 0.5$) | 0.586 | 0.4766 |
|  | ($AC_w = 0.6$) | 0.5866 | 0.4769 |
|  | ($AC_w = 0.7$) | 0.5864 | 0.4768 |
|  | ($AC_w = 0.8$) | 0.5864 | 0.4769 |
|  | ($AC_w = 0.9$) | 0.5857 | 0.4753 |

**Multiple Area Chairs**

We also experimented on a variant of ReFer where we have multiple ACs, and then have each of them give the scores with the ReFeR framework, but in the end, combine the scores by averaging them. In the results in Table 19 GPT AC rows refer to our original setup of GPT-4o-mini as our AreaChair model, and Claude AC refers to the Claude-3.5-haiku model as AC. We can observe that with the suggested method of 2 area chair ReFeR-Lite version, we achieve slightly more than the ReFeR-Lite method, whereas we can see that for both AC, the ReFeR-Pro gives significant gains, compared to the AC. These results demonstrate how the ReFeR method is effective even with multiple Area chairs.

Table 19: Average Spearman($\rho$) and Kendall-Tau($\tau$) correlation across metrics for different ACs and Multiple AC. The Multiple AC row refers to a ReFeR-Lite variant.

| Method | AC Model | Spearman ($\rho$) | Kendall-Tau ($\tau$) |
|---|---|---|---|
| ReFeR-Pro | GPT-4o-mini | 0.6352 | 0.5188 |
| ReFeR-Lite | GPT-4o-mini | 0.5827 | 0.5041 |
| ReFeR-Pro | Claude-3.5-haiku | 0.6593 | 0.5414 |
| ReFeR-Lite | Claude-3.5-haiku | 0.6325 | 0.5525 |
| Multiple AC | GPT + Claude | 0.6327 | 0.5378 |

**Method Ensemble**

We performed these experiments to observe if combining/ensembling other methods such as G-Eval has any improvement over the existing framework due to the possibility of the other method being complementary.

Table 20: Comparison of Average Spearman correlation for Methods' ensemble across 4 metrics.

| Method | Coherence | Engagingness | Groundedness | Naturalness | Average |
|---|---|---|---|---|---|
| G-Eval (n=20) | 0.596 | 0.503 | 0.549 | 0.616 | 0.566 |
| ReFeR-Pro | 0.589 | 0.689 | 0.637 | 0.626 | 0.635 |
| ReFeR-Lite | 0.520 | 0.669 | 0.600 | 0.575 | 0.591 |
| G-Eval + ReFeR-Pro | 0.609 | 0.630 | 0.635 | 0.634 | 0.627 |
| G-Eval + ReFeR-Lite | 0.593 | 0.639 | 0.628 | 0.624 | 0.621 |
| Weighted Ensemble | 0.601 | 0.543 | 0.576 | 0.635 | 0.589 |
| Inverse Weighted | 0.616 | 0.585 | 0.617 | 0.637 | 0.614 |

We ensemble by taking the average of the scores, in case of G-Eval and ReFeR-Pro, we average over the 20 + 20 scores, and in case of ReFeR-Lite, we just take the average of the ReFeR-Lite and G-Eval. In case of the weighted G-Eval, ReFeR-Pro Avg, we have chosen the weights using variance based confidence, basically assuming lower variance over 20 responses meaning higher confidence/weightage (This method is chosen because that is the only external factor to do some kind of ensembling other than simple average). From the results in Table 20 we can see that, for ReFeR-Pro, this ensemble deteriorates the performance, but for ReFeR-Lite, it slightly improves the performance. In the case of the weighted average, it reduced the performance drastically. But surprisingly in case of the inverse weighted avg, where higher variance was preferred, the performance reductions are not that high.

**ReFeR with Debate**

We performed experiments to see if including debate into the ReFeR framework has any improvement. And the results in Table 21 show that even though there are slight improvements over the normal framework with the debate in 2 metrics, overall, just the hierarchical structure without the Debate is better. We also see that the correlation of peers' scores drops drastically after debate. We have adopted the debate framework from the Multi-Agent Debate framework (Du et al., 2023) from our reasoning baselines, but where we have a single debate round and then the AC review.

Table 21: Comparison of ReFeR and ReFeR Debate variants across multiple evaluation dimensions.

| Category | Method | Coherence | Engagingness | Groundedness | Naturalness | Average |
|---|---|---|---|---|---|---|
| | Llama-3.1-8B | 0.312 | 0.451 | 0.395 | 0.330 | 0.372 |
| | Mistral Nemo | 0.441 | 0.623 | 0.429 | 0.374 | 0.467 |
| ReFeR | Gemma-2-9B | 0.505 | 0.603 | 0.582 | 0.516 | 0.552 |
| | ReFeR Pro | 0.589 | 0.689 | 0.637 | 0.626 | 0.635 |
| | ReFeR Lite | 0.537 | 0.630 | 0.583 | 0.565 | 0.579 |
| | Llama-3.1-8B | 0.211 | 0.158 | 0.130 | 0.224 | 0.181 |
| ReFeR | Mistral Nemo | 0.134 | 0.034 | 0.012 | 0.109 | 0.072 |
| + Debate | Gemma-2-9B | 0.466 | 0.449 | 0.436 | 0.433 | 0.446 |
| | ReFeR Pro | 0.603 | 0.634 | 0.650 | 0.586 | 0.618 |
| | ReFeR Lite | 0.514 | 0.552 | 0.586 | 0.537 | 0.547 |

# R   Prompts

## R.1   NLG Evaluation

**TopicalChat**

---

***Coherence Peer Prompt***

*You will receive a dialogue between two people. Following that, there will be one suggested reply for the next part of the conversation, along with a related interesting fact.*

*Your job is to assess how coherent the suggested reply is, focusing on its ability to seamlessly continue the dialogue while also considering the overall context of the conversation, including the provided fact.*

*Please read and understand these instructions carefully. You may refer back to them as needed.*

*Assessment Criteria:*

*Coherence (1-3): How well does the response continue the conversation?*

*- A score of 1 (no) indicates that the reply significantly shifts the topic or disregards the ongoing conversation entirely.*

---

- A score of 2 (somewhat) suggests that the response makes a vague reference to the conversation but fails to effectively engage with the dialogue or the accompanying fact.
- A score of 3 (yes) signifies that the response stays on topic, acknowledges the previous dialogue, and draws a clear and relevant connection to the interesting fact provided while maintaining the overall conversational flow.

Assessment Process:

1. Review the conversation history for context and flow, focusing on how well the suggested reply relates to the previous exchanges.

2. Examine the suggested reply for its relevance and engagement with the ongoing dialogue.

3. Consider how well the reply connects with the interesting fact while also evaluating its contribution to the conversation as a whole.

4. Assign a coherence score of 1, 2, or 3, taking into account both the conversational progression and the connection to the fact.

Example:
Conversation History: {{Conversation}}
Corresponding Fact: {{Contextual Fact}}
Response: {{Generated Response}}
Evaluation Form (Answer by starting with "Analysis:" to analyze the given example regarding the evaluation criteria as concise as possible, and then give the numeric rating on the next line by "Rating".)
Coherence:

---

**Coherence AreaChair Prompt**

Navigate through a simulated conversation between two individuals, followed by a provided potential response incorporating an intriguing fact. Your role is to assess the responses based on the coherence metric.

Alongside your evaluation, you will also receive initial evaluations from three large language models, referred to as the assistants' evaluations. Please read the instructions and criteria below carefully and use them as a guide in your evaluation, critically assessing the conversation, and the assistants' inputs.

Ensure a meticulous understanding of the instructions. Keep this document accessible for reference during the evaluation.

Evaluation Criteria:

Coherence (1-3): Assess whether the response seamlessly continues the conversation history.
- A score of 1 (no) denotes a significant shift in topic or disregard for the conversation history.
- A score of 2 (somewhat) indicates a response with limited reference to the conversation history and a noticeable shift in topic.
- A score of 3 (yes) signifies an on-topic response that strongly acknowledges and builds upon the conversation history.

Evaluation Steps:

1. Thoroughly read the conversation history.

2. Examine the potential response.

3. Evaluate coherence based on the conversation history.

*4. Assign a coherence score of 1, 2, or 3.*

*Example:*
*Conversation History: {{Conversation}}*
*Corresponding Fact: {{Contextual Fact}}*
*Response: {{Generated Response}}*
*First Assistant's Evaluation: {{Peer_response1}}*
*Second Assistant's Evaluation: {{Peer_response2}}*
*Third Assistant's Evaluation: {{Peer_response3}}*
*Evaluation Form (Answer by starting with "Analysis:" to analyze the given example regarding the evaluation criteria as concise as possible, and then give the numeric rating on the next line by "Rating".)*
*Coherence:*

---

**Engagingness Peer Prompt**

*You will be given a conversation between two individuals. You will then be given one potential response for the next turn in the conversation. The response concerns an interesting fact, which will be provided as well.*

*Your task is to rate the responses on one metric. Please make sure you read and understand these instructions carefully. Please keep this document open while reviewing, and refer to it as needed.*

*Evaluation Criteria:*

*Engagingness (1-3): Is the response dull/interesting?*
*- A score of 1 (dull) means that the response is generic and dull.*
*- A score of 2 (somewhat interesting) means the response is somewhat interesting and could engage you in the conversation (e.g., an opinion, thought).*
*- A score of 3 (interesting) means the response is very interesting or presents an interesting fact.*

*Evaluation Steps:*

*1. Read the conversation, the corresponding fact and the response carefully.*

*2. Rate the response on a scale of 1-3 for engagingness, according to the criteria above.*

*Example:*
*Conversation History: {{Conversation}}*
*Corresponding Fact: {{Contextual Fact}}*
*Response: {{Generated Response}}*
*Evaluation Form (Answer by starting with "Analysis:" to analyze the given example regarding the evaluation criteria as concise as possible, and then give the numeric rating on the next line by "Rating".)*
*Engagingness:*

---

**Engagingness AreaChair Prompt**

*Navigate through a simulated conversation between two individuals, followed by a provided potential response incorporating an intriguing fact. Your role is to assess the responses based on the engagingness metric.*

*Alongside your evaluation, you will also receive initial evaluations from three large language models, referred to as the assistants' evaluations. Please read the instructions and criteria below carefully and use them as a guide in your evaluation, critically assessing the conversation, and the assistants' inputs.*

*Ensure a meticulous understanding of the instructions. Keep this document accessible for reference during the evaluation.*

*Evaluation Criteria:*

*Engagingness (1-3): Is the response dull or interesting?*
*- A score of 1 (dull) means that the response is generic and uninteresting.*
*- A score of 2 (somewhat interesting) means the response is somewhat engaging and could capture interest (e.g., an opinion or thought).*
*- A score of 3 (interesting) means the response is highly engaging or presents an intriguing fact.*

*Evaluation Steps:*

1. *Read the conversation, the corresponding fact, and the response carefully.*

2. *Rate the response on a scale of 1-3 for engagingness, according to the criteria above.*

*Example:*

*Conversation History: {{Conversation}}*

*Corresponding Fact: {{Contextual Fact}}*

*Response: {{Generated Response}}*

*First Assistant's Evaluation: {{Peer_response1}}*

*Second Assistant's Evaluation: {{Peer_response2}}*

*Third Assistant's Evaluation: {{Peer_response3}}*

*Evaluation Form (Answer by starting with "Analysis:" to analyze the given example regarding the evaluation criteria as concisely as possible, and then give the numeric rating on the next line by "Rating".)*

*Engagingness:*

---

### Groundedness Peer Prompt

*You will be given a conversation between two individuals. You will then be given one potential response for the next turn in the conversation. The response concerns an interesting fact, which will be provided as well.*

*Your task is to rate the responses on one metric. Please make sure you read and understand these instructions carefully. Please keep this document open while reviewing, and refer to it as needed.*

*Evaluation Criteria:*

*Groundedness (0-1) given the fact that this response is conditioned on, determine whether this response uses that fact.*
*- A score of 0 (no) means the response does not mention or refer to the fact at all.*
*- A score of 1 (yes) means the response uses the fact well.*

*Evaluation Steps:*

1. *Read the conversation between the two individuals.*

2. *Identify the fact that is provided for the potential response.*

3. *Read the potential response.*

4. *Determine if the potential response uses or mentions the fact.*

5. *Assign a score of 0 or 1 for groundedness based on whether the response uses the fact.*

*Example:*

*Conversation History: {{Conversation}}*

*Corresponding Fact: {{Contextual Fact}}*

*Response: {{Generated Response}}*

*Evaluation Form (Answer by starting with "Analysis:" to analyze the given example regarding the evaluation criteria as concise as possible, and then give the numeric rating on the next line by "Rating".)*

*Groundedness:*

---

**Groundedness AreaChair Prompt**

*Navigate through a simulated conversation between two individuals, followed by a provided potential response incorporating an intriguing fact. Your role is to assess the responses based on the groundedness metric.*

*Alongside your evaluation, you will also receive initial evaluations from three large language models, referred to as the assistants' evaluations. Please read the instructions and criteria below carefully and use them as a guide in your evaluation, critically assessing the conversation, and the assistants' inputs. Ensure a meticulous understanding of the instructions. Keep this document accessible for reference during the evaluation.*

*Evaluation Criteria:*

*Groundedness (0-1): Given the fact that this response is conditioned on, determine whether this response uses that fact.*

*- A score of 0 (no) means the response does not mention or refer to the fact at all.*

*- A score of 1 (yes) means the response uses the fact well.*

*Evaluation Steps:*

1. *Read the conversation between the two individuals.*

2. *Identify the fact that is provided for the potential response.*

3. *Read the potential response.*

4. *Determine if the potential response uses or mentions the fact.*

5. *Assign a score of 0 or 1 for groundedness based on whether the response uses the fact.*

*Example:*

*Conversation History: {{Conversation}}*

*Corresponding Fact: {{Contextual Fact}}*

*Response: {{Generated Response}}*

*First Assistant's Evaluation: {{Peer_response1}}*

*Second Assistant's Evaluation: {{Peer_response2}}*

*Third Assistant's Evaluation: {{Peer_response3}}*

*Evaluation Form (Answer by starting with "Analysis:" to analyze the given example regarding the evaluation criteria as concisely as possible, and then give the numeric rating on the next line by "Rating".)*

*Groundedness:*

---

**Naturalness Peer Prompt**

*You will be given a conversation between two individuals. You will then be given one potential response for the next turn in the conversation. The response concerns an interesting fact, which will be provided as well.*

*Your task is to rate the responses on one metric. Please make sure you read and understand these instructions carefully. Please keep this document open while reviewing, and refer to it as needed.*

*Evaluation Criteria:*

*Naturalness (1-3) Is the response naturally written??*
*- A score of 1 (bad) means that the response is unnatural.*
*- A score of 2 (ok) means the response is strange, but not entirely unnatural.*
*- A score of 3 (good) means that the response is natural.*

*Evaluation Steps:*

    *1. Read the conversation between the two individuals.*

    *2. Read the potential response for the next turn in the conversation.*

    *3. Evaluate the response based on its naturalness, using the provided criteria.*

    *4. Assign a rating score of 1, 2, or 3 based on the evaluation.*

*Example:*

*Conversation History: {{Conversation}}*

*Corresponding Fact: {{Contextual Fact}}*

*Response: {{Generated Response}}*

*Evaluation Form (Answer by starting with "Analysis:" to analyze the given example regarding the evaluation criteria as concise as possible, and then give the numeric rating on the next line by "Rating".)*

*Naturalness:*

---

### Naturalness AreaChair Prompt

*Navigate through a simulated conversation between two individuals, followed by a provided potential response incorporating an intriguing fact. Your role is to assess the responses based on the naturalness metric.*

*Alongside your evaluation, you will also receive initial evaluations from three large language models, referred to as the assistants' evaluations. Please read the instructions and criteria below carefully and use them as a guide in your evaluation, critically assessing the conversation, and the assistants' inputs.*

*Ensure a meticulous understanding of the instructions. Keep this document accessible for reference during the evaluation.*

*Evaluation Criteria:*

*Naturalness (1-3): Is the response naturally written?*
*- A score of 1 (bad) means that the response is unnatural.*
*- A score of 2 (ok) means the response is strange, but not entirely unnatural.*
*- A score of 3 (good) means that the response is natural.*

*Evaluation Steps:*

    *1. Read the conversation between the two individuals.*

    *2. Read the potential response for the next turn in the conversation.*

    *3. Evaluate the response based on its naturalness, using the provided criteria.*

    *4. Assign a rating score of 1, 2, or 3 based on the evaluation.*

*Example:*

*Conversation History: {{Conversation}}*

*Corresponding Fact: {{Contextual Fact}}*

*Response: {{Generated Response}}*

*First Assistant's Evaluation: {{Peer_response1}}*

*Second Assistant's Evaluation: {{Peer_response2}}*

*Third Assistant's Evaluation: {{Peer_response3}}*

*Evaluation Form (Answer by starting with "Analysis:" to analyze the given example regarding the evaluation criteria as concisely as possible, and then give the numeric rating on the next line by "Rating".)*

*Naturalness:*

## R.2   MultiModal Evaluation

### R.2.1   ICQD

**Caption Quality Peer Prompt**

*Your task is to carefully evaluate the alignment between an image and its corresponding caption based on the provided criteria. Pay close attention to the instructions to ensure an accurate and nuanced assessment.*

*Instructions:*

1. *Examine the image closely, identifying its key visual elements, objects, actions, and overall context.*

2. *Scrutinize the caption, comparing it to the visual content of the image, and identifying any inaccuracies, omissions, or misleading information. Consider both the explicit details and the overall context of the image.*

3. *Rate the caption on a scale of 1-100 according to the Evaluation Criteria, where 1 indicates a very poor match and 100 indicates a perfect match.*

*Evaluation Criteria:*

*Rating (0-100): Evaluate the extent to which the caption aligns with the visual content of the image. A high rating should be given if the caption accurately reflects the main elements, actions, and context of the image, even if it uses concise language or omits minor details. Deduct points for inaccuracies, misleading descriptions, or significant omissions that distort the intended message of the image.*

*- 90-100: The caption perfectly or almost perfectly captures the image's content.*
*- 70-89: The caption is mostly accurate, with only minor inaccuracies or omissions.*
*- 50-69: The caption has notable inaccuracies or omissions but still partially represents the image.*
*- 30-49: The caption poorly represents the image, with significant inaccuracies or misleading elements.*
*- 0-29: The caption is almost entirely inaccurate or irrelevant to the image.*

*Example:*

*Image:*

*[Image will be provided separately]*

*Caption: {{Caption}}*

*Evaluation Form (Answer by starting with "Analysis:" to analyze the provided example regarding the evaluation criteria as concisely as possible, and then give the numeric rating on the next line by "Rating".)*

*Caption_Quality:*

---

***Caption Quality AreaChair Prompt***

*You will be given an image, its caption, and you will also receive initial evaluations from two large language models, referred to as the assistants' evaluations.*

*Your task is to rate the caption on one metric.*

*Please read the instructions and criteria below carefully and use them as a guide in your evaluation.*

*Evaluation Criteria:*

*Relevance (0-100) - Assess how well the caption aligns with the content of the image. The caption should accurately describe or complement the visual elements and context of the image. Consider if the caption captures the key aspects of the image, its mood, and its intent, and whether it adds value by enhancing the viewer's understanding or experience of the image.*

*Evaluation Guidelines:*

1. *Examine the Image: Carefully observe the image to understand its main elements, context, and message.*
2. *Review the Caption: Analyze if the caption accurately and effectively describes or complements the image. Consider the appropriateness of the language, tone, and whether the caption adds meaningful context or insight.*
3. *Rate the Caption's Relevance on a Scale of 0 to 100:*
   - *90-100: The caption is highly relevant, fully capturing the essence of the image with precise and insightful description or commentary, adding significant value to the image.*
   - *80-89: The caption is mostly relevant, capturing most key elements of the image with minor omissions or slightly less impactful language, still adding clear value.*
   - *70-79: The caption is somewhat relevant, capturing some key aspects but missing others, or includes minor irrelevant details, with a noticeable but limited enhancement to the image.*
   - *50-69: The caption has limited relevance, covering only a few elements of the image or providing a description that is either too generic or somewhat off-target, adding minimal value.*
   - *30-49: The caption is marginally relevant, with significant omissions or inaccuracies, possibly detracting from the image by misrepresenting it or providing little to no useful context.*
   - *10-29: The caption is largely irrelevant, missing the key aspects of the image, with significant inaccuracies or misrepresentations, adding no value or even confusing the viewer.*
   - *0-9: The caption is completely irrelevant or nonsensical, with no connection to the image, possibly confusing or misleading the viewer.*

*Example:*

*Image:*

*[Image is attached below]*

*Caption: {{Caption}}*

*First Assistant's Evaluation: {{Peer_Response1}}*

*Second Assistant's Evaluation: {{Peer_Response2}}*

*Evaluation Form (Answer by starting with "Analysis:" to analyze the provided example regarding the evaluation criteria, incorporating the peer ratings, and then give the numeric rating on the next line by "Rating".)*

*Caption_Quality:*

## R.2.2 AGIQA

***Image Quality Peer Prompt***

*You will be given an image generated based on an input prompt.*

*Your task is to rate the image on one metric.*

*Please make sure you read and understand these instructions carefully. Please keep this document open while reviewing, and refer to it as needed.*

*Evaluation Criteria:*

*Image_Quality (0-5) - the overall visual coherence and alignment with the input prompt. This rating should reflect how well the image matches the prompt, considering the clarity, relevance, and composition of the image.*

*Evaluation Steps:*

- *Review the "Input Prompt" carefully to understand the intended content, theme, and style.*
- *Examine the generated image and compare it to the "Input Prompt". Check if the image accurately represents the prompt, is visually clear, and if the composition aligns with the expected outcome.*
- *Assign a score for Image Quality on a scale of 0 to 5, where 0 is the lowest and 5 is the highest based on the Evaluation Criteria.*

*Example:*

*Input Prompt: {{Input_Prompt}}*

*Generated Image:*

*[Image is attached below]*

*Evaluation Form (Answer by starting with "Analysis:" to analyze the given example regarding the evaluation criteria as concise as possible, and then give the numeric rating on the next line by "Rating".)*

*- Image_Quality:*

---

### Image Quality AreaChair Prompt

*You will be given an image generated based on an input prompt, along with initial evaluations from two assistants, referred to as the assistants' evaluations.*

*Your task is to rate the image on one metric.*

*Please read the instructions and criteria below carefully and use them as a guide in your evaluation.*

*Evaluation Criteria:*

*Image_Quality (0-5) - Assess the visual coherence and alignment of the image with the input prompt. The image should reflect the content, theme, and style described in the prompt, and be visually clear and well-composed.*

*Evaluation Guidelines:*

- *Review the "Input Prompt" to understand the intended content, theme, and style.*
- *Examine the generated image and analyze how well it represents the "Input Prompt" in terms of accuracy, clarity, and composition.*
- *Rate the image's quality on a scale of 0 to 5, with 0 being the lowest quality and 5 being the highest quality.*
- *Scoring Guidelines:*
  - *Score 5.0: The image fully captures the essence of the prompt with a high level of accuracy, clarity, and visual appeal, without any significant errors or irrelevant elements.*
  - *4 ≤ Score < 5: The image mostly aligns with the prompt, with minor inaccuracies or less relevant details, but still maintains a generally high quality.*
  - *3 ≤ Score < 4: The image partially represents the prompt, with noticeable inaccuracies or irrelevant details, and a less coherent visual presentation.*
  - *2 ≤ Score < 3: The image has significant deviations from the prompt, with major inaccuracies, irrelevant elements, and a disjointed visual composition.*
  - *1 ≤ Score < 2: The image fails to represent the prompt accurately, lacks visual coherence, and includes significant errors or irrelevant elements.*

*− 0 ≤ Score < 1: The image is completely unrelated to the prompt.*

*Example:*

*Input Prompt: {{Input_Prompt}}*

*Generated Image:*

*[Image is attached below]*

*First Assistant's Evaluation: {{Peer_response1}}*

*Second Assistant's Evaluation: {{Peer_response2}}*

*Please provide your analysis and rating as follows:*

*Evaluation Form (Answer by starting with "Analysis:" to analyze the given example regarding the evaluation criteria as concise as possible, and then give the numeric rating on the next line by "Rating".)*

*- Image_Quality:*

### R.3 Reasoning

### R.3.1 AQuA

**AQuA Peer Prompt**

*You will be provided with a problem that requires logical reasoning, mathematical calculation, or both.*

*Your task is to solve the problem accurately, providing not just the correct answer but also a clear explanation of the steps taken to reach that answer.*

*It is crucial to thoroughly understand the problem and apply the correct principles or formulas to solve it.*

*Instructions:*

- *Read the problem statement carefully, ensuring you understand all the details and what is required for the solution.*
- *Work through the problem logically and methodically, explaining your reasoning and the steps you take to solve the problem.*
- *Provide the final answer clearly, specifying it by choosing one of the provided options (e.g., A, B, C, etc.).*

*Problem Statement: {{question}}*

*Provided Options: {{options}}*

*Evaluation Form:*

- *Analysis: [Start with "Analysis:" to provide a concise and structured explanation of the steps and reasoning used to solve the problem. Ensure your analysis is clear and follows a logical sequence.]*
- *Answer: [Clearly state the final answer only (e.g., A, B, C, etc.) on the line after your analysis.]*

**AQuA area chair Prompt**

*You will be provided with a problem that requires logical reasoning, mathematical calculation, or both. Along with the problem, you will also receive solutions from three other Language Models (LLMs).*

*Your task is to solve the problem accurately, using the peer responses to inform your approach. Apply the correct principles or formulas to arrive at the solution, while taking note of any useful insights or errors in the peer responses.*

*Instructions:*

- *Understand the Problem: Read the problem statement carefully, ensuring you grasp all details.*

- *Review Peer Responses: Consider the solutions provided by the LLMs, noting useful approaches or any errors.*
- *Solve the Problem: Work through the problem logically, explaining your reasoning and steps. Utilize the peer responses as needed but ensure your solution is accurate and complete.*
- *Final Answer: Clearly state the final answer, choosing one of the provided options (e.g., A, B, C, etc.).*

*Problem Statement: {{question}}*

*Provided Options: {{options}}*

*Solutions by Other LLMs:*

- *LLM 1 Answer: {{Peer_response1}}*
- *LLM 2 Answer: {{Peer_response2}}*
- *LLM 3 Answer: {{Peer_response3}}*

*Evaluation Form:*

- *Analysis: [Start with "Analysis:", provide a concise explanation of your reasoning and steps, integrating relevant insights from the LLMs' responses.]*
- *Answer: [Clearly state the final answer label ONLY (e.g., A, B, C, etc.) on the line after your analysis. (DO NOT GIVE ANYTHING ELSE).]*

### R.3.2   BBH_DU

**BBH_DU Peer Prompt**

*You will be provided with a problem that requires understanding and interpreting dates or times logically.*

*Your task is to solve the problem accurately, providing not just the correct answer but also a clear explanation of the steps taken to reach that answer.*

*It is crucial to thoroughly understand the problem, applying the correct principles or formulas to arrive at the solution.*

*Instructions:*

- *Read the problem statement carefully, ensuring you understand all the details and what is required for the solution.*
- *Work through the problem logically and methodically, explaining your reasoning and the steps you take to solve the problem.*
- *Provide the final answer clearly, specifying it by choosing one of the provided options (e.g., A, B, C, etc.).*

*Problem Statement: {{question}}*

*Evaluation Form:*

- *Analysis: [Start with "Analysis:" to provide a concise and structured explanation of the steps and reasoning used to solve the problem. Ensure your analysis is clear and follows a logical sequence.]*
- *Answer: [Clearly state the final answer only (e.g., A, B, C, etc.) on the line after your analysis.]*

**BBH_DU AreaChair Prompt**

*You will be provided with a problem that requires understanding and interpreting dates or times logically. You will also receive the final answers from three other Language Models (LLMs).*

*Your task is to solve the problem accurately, using the answers provided by the LLMs to inform your reasoning. Provide a clear explanation of your approach, and arrive at your own final answer.*

*Instructions:*

- *Understand the Problem: Read the problem statement carefully, ensuring you grasp all details.*
- *Review Peer answers: Consider the final answers provided by the LLMs, noting any patterns or outliers.*
- *Solve the Problem: Work through the problem logically, explaining your reasoning and steps. Use the peer answers as a reference but ensure your solution is accurate and complete.*
- *Final Answer: Clearly state the final answer, choosing one of the provided options (e.g., A, B, C, etc.).*

*Problem Statement: {{question}}*

*Answers from Other LLMs:*

- *LLM 1 answer: {{Peer_response1}}*
- *LLM 2 answer: {{Peer_response2}}*
- *LLM 3 answer: {{Peer_response3}}*

*Evaluation Form:*

- *Analysis: [Start with "Analysis:" to provide a concise explanation of your reasoning and steps to solve the problem, using the peer answers as a reference.]*
- *Answer: [Clearly state the final answer label ONLY (e.g., A, B, C, etc.) on the line after your analysis. (DO NOT GIVE ANYTHING ELSE).]*

### R.3.3  CSQA

**CSQA Peer Prompt**

*Evaluate the question by selecting the best option from the provided choices. Your task is to understand the context and nuances of the question, utilize your knowledge of the topic, and determine the most appropriate answer based on the options given. The goal is to select the most relevant and correct option that aligns with the question's intent.*

*Instructions:*

- *Understand the Question: Read the question carefully to comprehend all aspects and the context in which it is asked.*
- *Consider the Options: Analyze each provided option carefully. Think about how each option relates to the question and the scenario it presents.*
- *Select the Best Option: Choose the option that best answers the question, based on your analysis. Focus on the logic or knowledge that supports this choice.*

*Problem Statement: {{question}}*
*Provided Options: {{options}}*
*Evaluation Form:*

- *Analysis: [Begin with "Analysis:" to provide a structured and clear explanation of your reasoning process. Your analysis should logically explain why the chosen option is the most appropriate answer to the question.]*
- *Answer: [Clearly state the final answer only (e.g., A, B, C, etc.) on the line after your analysis.]*

---

**CSQA area chair Prompt**

*You will be provided with a question that requires careful evaluation to select the best option from the provided choices. You will also receive the final answers from three other Language Models (LLMs).*

*Your task is to determine the most appropriate answer, using the answers provided by the LLMs to inform your reasoning. Provide a clear explanation of your thought process and select the option that best aligns with the question's intent.*

*Instructions:*

- *Understand the Question: Read the question carefully to comprehend all aspects and context.*
- *Review Peer Answers: Consider the final answers provided by the LLMs, noting any patterns or outliers.*
- *Select the Best Option: Based on your understanding and the peer answers, choose the option that best answers the question.*
- *Final Answer: Clearly state the final answer, choosing one of the provided options (e.g., A, B, C, etc.).*

*Problem Statement: {{question}}*

*Provided Options: {{options}}*

*Answers from Other LLMs:*

- *LLM 1 Answer: {{Peer_response1}}*
- *LLM 2 Answer: {{Peer_response2}}*
- *LLM 3 Answer: {{Peer_response3}}*

*Evaluation Form:*

- *Analysis: [Start with "Analysis:" to provide a concise and clear explanation of your reasoning, using the peer answers as a reference.]*
- *Answer: [Clearly state the final answer label ONLY (e.g., A, B, C, etc.) on the line after your analysis. (DO NOT GIVE ANYTHING ELSE).]*

### R.3.4 GSM8k

**GSM8k Peer Prompt**

*You will be provided with a problem that requires logical reasoning, mathematical calculation, or both.*

*Your task is to solve the problem accurately, providing not just the answer but also a clear explanation of the steps taken to reach that answer.*

*It is crucial to understand the problem thoroughly and apply the correct principles or formulas to solve it.*

*Instructions:*

- *Read the problem statement carefully, ensuring you understand all the details and what is required for the solution.*
- *Work through the problem logically and methodically, explaining your reasoning and the steps you take to solve the problem.*
- *Provide the final answer clearly, specifying it as a numerical value or a specific explanation as required by the problem statement.*

*Problem Statement: {{question}}*

*Evaluation Form:*

- *Analysis: [Start with "Analysis:" to provide a concise and structured explanation of the steps and reasoning used to solve the problem. Ensure your analysis is clear and follows a logical sequence.]*
- *Answer: [Clearly state the final answer only (number) on the line after your analysis.]*

---

**GSM8k area chair Prompt**

*You will be provided with a problem that requires logical reasoning, mathematical calculation, or both. You will also receive the final answers from three other Language Models (LLMs).*

*Your task is to solve the problem accurately, using the peer answers to inform your reasoning. Provide a clear explanation of your thought process and the steps taken to arrive at the solution. Ensure that your reasoning is sound and the final answer is correct.*

*Instructions:*

- *Understand the Problem: Read the problem statement carefully to ensure you grasp all the details and what is required.*
- *Review Peer Answers: Consider the final answers provided by the LLMs, noting any patterns or outliers.*
- *Work Through the Problem: Solve the problem methodically, using the peer answers as a reference. Explain your reasoning clearly.*
- *Final Answer: Provide the final answer clearly, specifying it as a numerical value or as required by the problem statement.*

*Problem Statement: {{question}}*

*Answers from Other LLMs:*

- *LLM 1 Answer: {{Peer_response1}}*
- *LLM 2 Answer: {{Peer_response2}}*
- *LLM 3 Answer: {{Peer_response3}}*

*Evaluation Form:*

- *Analysis: [Start with "Analysis:" to provide a concise and clear explanation of your reasoning, using the peer answers as a reference.]*
- *Answer: [Clearly state the final answer ONLY (number) on the line after your analysis. (DO NOT GIVE ANYTHING ELSE).]*

