# OpenReview forum: "ReFeR: Improving Evaluation and Reasoning through Hierarchy of Models"
_TMLR — Accepted by TMLR_

### Review · Reviewer_EWhs · 2025-05-27

**Summary Of Contributions:**

This paper introduces ReFeR, a hierarchical multi-agent framework for evaluating generative outputs (text and images) and enhancing reasoning tasks. ReFeR employs a two-tiered structure: multiple "peer" models (e.g., smaller LLMs/VLMs) independently evaluate outputs, and a more capable "area chair" model synthesizes their feedback to produce final scores and explanations. The framework offers two variants: ReFeR-Turbo (high accuracy, higher compute) and ReFeR-Lite (compute-efficient, comparable performance). Experiments across four NLG/multimodal evaluation benchmarks and four reasoning tasks demonstrate ReFeR’s superiority over single-model and multi-agent baselines. The authors also highlight its utility for generating feedback-rich datasets to distill evaluation capabilities into smaller models.

**Audience:**

Yes

**Claims And Evidence:**

Yes

**Requested Changes:**

- Missing discussion of some recent multi-agent reasoning works in related work and ensemble learning approaches in NLP.
- How does performance scale with number of peers beyond the tested range? The diminishing returns observed suggest optimal peer counts may vary by task.
- The finding that "scores only" works best is interesting but counterintuitive - this deserves deeper investigation.

**Strengths And Weaknesses:**

# Strengths
- **Well-motivated framework design**: The academic peer review analogy is intuitive and provides clear structure for the multi-agent approach. The hierarchical design with specialized roles is conceptually sound.
- **Comprehensive evaluation**: The paper demonstrates effectiveness across diverse tasks (NLG evaluation, multimodal assessment, reasoning) with consistent improvements over baselines.
- **Thorough analysis**: The ablation studies on peer selection, communication strategies, and error analysis provide valuable insights into framework behavior.

# Weaknesses
- **Limited technical novelty**: While the peer review framing is intuitive, the core contribution is essentially ensemble learning with LLMs. The technical advances beyond existing multi-agent approaches are incremental.
- **Questionable baseline comparisons**: For multimodal tasks, comparing against training-free methods while excluding trained models like ImageReward/PickScore creates unfair comparisons; The reasoning baselines are somewhat limited - missing recent strong multi-agent reasoning methods
- **Cost-benefit analysis unclear**: ReFeR-Turbo uses 4 models vs single model baselines, making cost comparisons misleading; The "12-14× more efficient" claim for ReFeR-Lite needs clearer justification; Table 2 shows ReFeR-Turbo as most expensive (1.0 relative cost) but claims about efficiency are confusing

---

> ### Author Response · Authors · 2025-06-15
> **Rebuttal (1/2)**
>
> # **Weaknesses:**
>
> > Limited technical novelty: While the peer review framing is intuitive, the core contribution is essentially ensemble learning with LLMs. The technical advances beyond existing multi-agent approaches are incremental.
>
> We appreciate the reviewer's point and agree that ReFeR can indeed be seen as a form of ensembling. However, we believe its novelty lies in the structured and purposeful way the ensemble is designed, which sets it apart from simple averaging or unstructured debate-based approaches.
>
> Unlike standard ensemble methods where all models are equal, ReFeR introduces a clear hierarchy within the models. It uses multiple diverse, lower-cost models as "Peers" and a single, higher-capability model as the "Area Chair." This setup is our choice of architecture for combining the strengths of different models more effectively.
>
> Moreover, the Area Chair’s role goes beyond just averaging peer scores. It is explicitly prompted to synthesize the feedback from peers—resolve disagreements, weigh inputs, and give a final evaluation/reasoning that is more nuanced than what a simple vote or average could provide.
>
> Another key contribution is our prompting design. We introduce a novel prompting schema tailored for this hierarchical structure, including an "Auto Prompt" method (detailed in Appendix E) that generates evaluation guidelines dynamically. This structured prompting schema as shown in Appendix D (Prompt Ablation) where we clearly see how the evaluation guidelines schema improves model performance.
>
> > Questionable baseline comparisons: For multimodal tasks, comparing against training-free methods while excluding trained models like ImageReward/PickScore creates unfair comparisons; The reasoning baselines are somewhat limited - missing recent strong multi-agent reasoning methods.
>
> *Multimodal Baselines:* The reviewer is correct that trained models like ImageReward and PickScore are very strong. But our primary goal was to advance the state-of-the-art for tuning-free evaluation frameworks, which can be applied "off-the-shelf" without requiring task-specific training data. We included these trained models in Table 3 to compare our performance against the SOTA, but our main claim is superiority within the tuning-free frameworks. Our ReFeR-Lite variant surpasses the other tuning-free VLM method (X-IQE) by a large margin and is competitive with the trained models, which we see as a significant result.
>
> *Reasoning Baselines:* Many multi-agent frameworks have been developed in recent times. Our baselines (Multi-Agent Debate, Multi-Agent Peer Review) were chosen as they are prominent and well-established representatives of same-model multi-agent frameworks, allowing for a fair comparison. We explained in Appendix H on why we can’t compare with frameworks such as Reconcile. More recent works were not included as they are contemporary to our paper and due to codebases not being available for few works.
>
> > Cost-benefit analysis unclear: ReFeR-Turbo uses 4 models vs single model baselines, making cost comparisons misleading; The "12-14× more efficient" claim for ReFeR-Lite needs clearer justification; Table 2 shows ReFeR-Turbo as most expensive (1.0 relative cost) but claims about efficiency are confusing.
>
> Regarding the cost of ReFeR vs. Baselines, the "single model baselines" (G-Eval, Analyze-Rate) are not single-call methods. They rely on generating n=20 responses from a powerful model (GPT-4o-mini in our setup) to achieve their reported performance.
> Therefore, the cost comparison is not 4 models vs. 1 model, but rather (3 cheap peer calls + 20 expensive AC calls) for ReFeR-Turbo vs. (20 expensive model calls) for the baselines. And more over our ReFeR-Lite is made of 3 cheap peer calls and a single expensive AC Call which is compared to the baselines which are made of 20 expensive calls.
>
> Hence the "12-14x more efficient" claim, it is exclusively for ReFeR-Lite. Since the cost of the cheap, open-source peers is negligible (or zero if self-hosted) compared to the expensive proprietary AC model. We aim to show how we can achieve better correlation/accuracy with few/single calls to the stronger, costlier models (ex:GPT-4o-mini) while leveraging the peer models and the hierarchical structure of evaluation.
>
> Regarding ReFeR-Turbo, it is our highest-performance, highest-cost variant. It is designed to maximize accuracy, not efficiency. The efficiency claims in the paper apply to ReFeR-Lite.

---

> > ### Author Response · Authors · 2025-06-15
> > **Rebuttal (2/2)**
> >
> > # **Requested Changes:**
> >
> > > 1. Missing discussion of some recent multi-agent reasoning works in related work and ensemble learning approaches in NLP.
> >
> > We haven’t initially addressed some of the recent works in the field of multi-agent reasoning because most of those works were contemporary. To address this, we have now added another section in the appendix, discussing these newer works in multi-agent reasoning, in the camera-ready version. We discuss alternative architectures, frameworks like CONSENSAGENT, which use flat, iterative debate structures to reach group consensus rather than relying on a final synthesizer. The DOWN(Debate Only When Necessary) framework, which dynamically activates debate based on confidence scores. Role-Playing and Knowledge-Enhanced Systems: like CoMM and MADKE, which explore specialized agent roles and the integration of external knowledge, respectively, highlighting different dimensions of multi-agent collaboration. Please refer to Section I of Appendix in the updated draft for detailed discussion.
> >
> > > 2. How does performance scale with number of peers beyond the tested range? The diminishing returns observed suggest optimal peer counts may vary by task.
> >
> > We already did the performance scaling with 5 peers in table-6 and figure-2 and it is practically difficult for us to do it again with more than 5 peers to show further graph of the performance. If we are to use more than 5 peers, we will again fall into the loop of trying to find peers of similar ability and not confuse ACs. Even in the real world setting we never review a paper with more than 5-6 reviewers and hence we leave this challenge to the future works. We acknowledge that the optimal number of peers may vary from task to task, but based on our ablation experiments with varying numbers of peers, it is likely that similar trends will be shown for various tasks where the framework performance stagnates/ doesn’t improve much after ~3-4 peers.
> >
> > Hence, based on the trade-off of compute-performance improvement, and considering real world settings where we may not use a large number of peers, we proposed a 3 peer framework. Hence one future direction of work can focus on finding the optimal number of models in a framework based on domain(Evaluation,reasoning, math etc) or other factors.
> >
> > > 3. The finding that "scores only" works best is interesting but counterintuitive - this deserves deeper investigation.
> >
> > We agree that the method of scores only being the best was counter-intuitive. We hypothesize that this is likely due to the long context leading to hallucination in the model. It is even more clear when passing both comments and scores, leading to the model getting confused more than the remaining methods. We hypothesize that providing only scores creates a cleaner, more direct signal for the Area Chair, reducing confusion from potentially conflicting or verbose peer analyses.

---

### Review · Reviewer_mdx1 · 2025-06-03

**Summary Of Contributions:**

The paper introduces ReFeR (Reason-Feedback-Review), a hierarchical multi-agent framework for evaluating generative model outputs. The framework uses smaller language models as "peer reviewers" and a larger model as an "area chair" to evaluate text and image generation quality. The authors present two variants: ReFeR-Turbo (using n=20 responses from the area chair) and ReFeR-Lite (n=1). The main contributions are: (1) A hierarchical evaluation framework inspired by academic peer review, (2) Evaluation on NLG tasks (TopicalChat, SummEval), multimodal tasks (ICQD, AGIQA), and reasoning benchmarks

**Audience:**

Yes

**Broader Impact Concerns:**

not applicable.

**Claims And Evidence:**

Yes

**Requested Changes:**

Provide deeper theoretical or methodological insights beyond the empirical setup. What fundamental principles make this hierarchical approach work better?
Include analysis of output consistency across multiple runs - in production settings, consistent scoring is crucial
Characterize what specific expertise or perspectives the peer reviewers contribute that the area chair (GPT-4o-mini) lacks. The accuracy boost alone doesn't capture this important aspect

**Strengths And Weaknesses:**

Strengths:

Comprehensive evaluation across multiple domains (NLG, multimodal, reasoning)
Strong ablation studies examining peer selection, number of peers, and area chair choice
Detailed analysis including FLOPs vs accuracy comparison
Well-documented implementation details and prompting strategies

Weaknesses:

The "inference time compute" framing appears superficial - the approach essentially involves calling multiple APIs, which is not a novel contribution
The comparison between ReFeR-Lite (n=1) and baselines (n=20) is somewhat obvious; claiming "12-14x compute efficiency" provides limited additional insight
The framework represents primarily an engineering and empirical setup rather than offering fundamentally new knowledge to the community
The paper focuses heavily on accuracy improvements but doesn't explore other important production metrics like output consistency
Missing analysis on how peer reviewers complement the area chair's capabilities - are they covering different aspects/expertise or just adding another call (for instance, what if we just replace 1 call to peer with another gpt4o-mini as a peer? ). It would otherwise not been convincing.

---

> ### Author Response · Authors · 2025-06-15
> **Rebuttal**
>
> >The "inference time compute" framing appears superficial - the approach essentially involves calling multiple APIs, which is not a novel contribution.
>
> We agree with the reviewer that "calling multiple APIs" is not, in itself, a novel contribution. We apologize if our "inference time compute" framing overshadowed the core methodological novelty of our work. We will rephrase wherever we mentioned it as a unique contribution of ours.
> The primary contribution of ReFeR is not simply using multiple models, but proposing a structured, hierarchical, and principled framework for orchestrating them. Wherever we have mentioned inference time compute, our goal was to show the effectiveness of the frameworks’ evaluation capabilities without any training. Using multiple APIs for models is just an engineering choice we took for the ease of using these models without actually deploying them on GPUs.
>
> > The comparison between ReFeR-Lite (n=1) and baselines (n=20) is somewhat obvious; claiming "12-14x compute efficiency" provides limited additional insight.
>
> We understand that the comparison is obvious, we just wanted to highlight that part to show how efficient this framework can be. We claim the 12-14x compute efficiency, in comparison with previous baselines like G-Eval and Analyze-Rate to show that our method will be able to get better results than the baselines even without resorting to the n=20 sampling which the previous works have proven to be a method to improve correlation. The compute efficiency claim is meant to underscore that ReFeR offers a practical path to high-quality automated evaluation in resource-constrained settings where running a model 20 times per sample is infeasible.
>
> > The framework represents primarily an engineering and empirical setup rather than offering fundamentally new knowledge. The paper focuses heavily on accuracy improvements but doesn't explore other important production metrics like output consistency.
>
> The goal of ReFeR is to give better correlation with humans at evaluating natural language questions while also improving the overall reasoning of the group. And hence we focus primarily only on accuracy and correlation rather than output consistency. And all our experiments were done on 3 runs and results were averaged across runs to maintain consistency. For eg: Table-4, this shows that our ReFeR-Turbo has one of the least deviations and even ReFeR-Lite is very much stable given only that we do a single final call for the answer.
>
> > Missing analysis on how peer reviewers complement the area chair's capabilities... It would otherwise not be convincing.
>
> We showed an example in appendix N which shows how the framework helps the reasoning task (gsm8k) to reconsider its solution by using the clues left in the peers' solution. We will add more details in the camera ready to help users understand how the peers are complimenting area chairs capabilities.

---

### Review · Reviewer_SJ6p · 2025-06-03

**Summary Of Contributions:**

This paper introduces ReFeR, a new method for conducting reference-free evaluation of LLMs & VLMs. To that end, the proposed approach leverages multiple **evaluator** LLMs in a setting that resembles peer review, where a few of the evaluator LLMs act as "peer reviewers" that each assign a score & a comment of the evaluated output. These scores & comments are then sent to an "area chair" evaluator LLM, which takes into account and adjudicates the peer reviewers' scores & comments, and then outputs a final score & comment for the evaluated output. To reduce variance & improve performance further, the more powerful version of ReFeR, denoted as ReFeR-Turbo, takes multiple outputs (up to n=20) from the "area chair" LLM, and produces a final score by doing a simple averaging of the area chair LLM's generated scores.

Experiments on various NLG, multimodal, and reasoning benchmarks demonstrate that the proposed approach (ReFeR-Turbo & ReFeR-Lite) generally outperforms other evaluator LLM baselines like G-Eval & LLM-as-Judge (for NLG evaluations) and zero-shot CoT (Kojima et al., 2023) for reasoning, and also outperforms each of the individual area chair / reviewer model performance. The paper then proceeds to conduct various ablations, concluding that: (i) going beyond 3 peer reviewer evaluator LLMs consisting of strong models provides only marginal gains; (ii) the strongest model should be the area chair, and that a group of capable peers should be chosen based on their individual performance; and (iii) the area chair has a higher probability of producing the correct answer if at least one of the peer reviews is correct, demonstrating that the area chair evaluator LLM *does* take into account the peer reviewer evaluator LLMs' output into account, although this effect is weaker for the reasoning tasks, suggesting that the area chair LLM uses more of its own knowledge for these types of tasks.

**Audience:**

Yes

**Broader Impact Concerns:**

No broader impact concerns come to mind.

**Claims And Evidence:**

Yes

**Requested Changes:**

1. **Critical**: Elaborate & analyze on why the G-Eval baseline is performing slightly better than ReFeR-Turbo in terms of Kendall-Tau correlation in Table 2. (Weakness 1)

2. **Critical**: Add another baseline that averages the 3 peer reviewer scores, plus the area chair score (i.e. average the scores from 4 models, rather than 3), potentially, with a larger weighting for the area chair model which is the strongest one. (Weakness 2)

3. **Strongly recommended**: Address the comments, suggestions, potential extensions, and typos on the suggestions section above.

**Strengths And Weaknesses:**

# Strengths
1. The question of how we can evaluate LLM outputs, including in cases where there is no reference text, is an increasingly important and timely one. As human evaluations are costly, using evaluator LLMs to judge the outputs of other LLMs is an important & active research area, where the proposed approach improves upon the state of the art.

2. The proposed approach is simple, intuitive, effective, and easy to implement, which increases the likelihood of being adopted more broadly in the field.

3. The paper features evaluations on multiple tasks & benchmarks, in addition to thorough ablations. This improves the credibility of the findings, and the ablations are useful to better understand and pinpoint the optimal hyper-parameters (how many peer reviewers, which model to use as the area chair, etc.) for using the proposed approach.

# Weaknesses
1. Results on the Topical Chat dataset are slightly worse than the G-Eval baseline in terms of the average Kendall-Tau correlations (Table 2, "Average" column), even though the ReFeR Turbo & Lite results are markedly better than the baseline in terms of the Spearman correlation. It'd be nice to understand & analyze why this is the case.

2. While there is a baseline that simply averages the 3 peer review evaluator LLMs (e.g. Table 2, "Peer Average" row), this is not exactly a fair comparison, because the ReFeR approach uses 4 models (3 peer reviewers + 1 area chair), rather than 3. It would be good to add a baseline that averages all 4 models, possibly with a higher weighting for the area chair model, which is the strongest one. This would help ascertain that the ReFeR approach is doing more than a simple averaging.

3. I still have a lot of minor comments regarding the terminology, typos, presentations, and potential extensions. These are listed under the "Suggestions / Questions / Typos" section below.

# Suggestions / Questions / Typos
1. This is a minor terminology point, but I am a little ambivalent about the term "ReFeR-Turbo". The reason is that the word Turbo is typically associated with increased speed, which I don't think is what the authors mean (there is a ReFeR-Light version for increased speed / less computational load). Maybe "Pro" or something like that?

2. Rather than taking n=20 samples from the area chair model (as is the case for ReFeR-Turbo), which is problematic because: (i) many of the models do not support taking 20 samples, and (ii) the samples are obtained from the same model, which means that they are likely to be correlated, it would be great to see results with multiple (e.g. 2) area chair models. Given that using n=1 (ReFeR-Lite) is not much worse than ReFeR-Turbo, I think using n=1 from 2 different, held-out area chair models could potentially yield better / competitive results than using n=20 from the same area chair model.

3. Rather than using manually defined prompts to create the reviewers & area chair model, would using automated prompt discovery methods (e.g. prompt tuning & prefix tuning) and / or parameter-efficient fine-tuning (e.g. LoRA, adapters, etc.) yield better peer reviewers & area chair models?

4. Could the peer reviewers / area chair model generate out-of-range scores (e.g. "6" while the range is 1-5)? If so, what happens in that case?

5. How much worse would the result be if ReFeR uses different personas of the same model for the peer reviewers, rather than 3 different models?

6. Typo: On page 6, "... comparable performance which being significantly cheaper". **which** should be replaced by **while**.

7. Would it be possible to ensemble different metrics, e.g. G-Eval & ReFeR Turbo / Lite? It would be great if we can get better results with ensembling, which suggests that the metrics are complementary.

8. Typo: On page 8, "... can result in performance better than the base ....". **performance better** -> **better performance**.

9. In Table 7, the row below GPT-4o-mini just says "ReFeR". This is ambiguous; do the authors mean "ReFeR-Turbo"?

10. Would the results be improved if the peer reviewers first do debating, and then revise their scores accordingly? Then the area chair evaluator LLM can take these post-debate peer reviewers' output, and then synthesize & adjudicate those instead of the pre-debate ones.

---

> ### Author Response · Authors · 2025-06-15
> **Rebuttal (1/3)**
>
> We thank the reviewer for the detailed feedback and review. We have addressed the weaknesses and suggestions below.
>
> **Weaknesses:**
>
> >1.  Results on the Topical Chat dataset are slightly worse than the G-Eval baseline in terms of the average Kendall-Tau correlations (Table 2, "Average" column), even though the ReFeR Turbo & Lite results are markedly better than the baseline in terms of the Spearman correlation. It'd be nice to understand & analyze why this is the case.
>
> This is due to the way ties are handled in the kendall-tau correlation coefficient. We have addressed the same issue in the summeval dataset results in Appendix F, where we have explained how a large number of ties have led to a lower kendall-tau correlation coefficient. In topical chat, even though the distribution isn’t that skewed, there are still lots of ties between samples, leading to the kendall-tau being a bit ambiguous at times, and hence not being a perfect metric for comparison.
>
> > 2. It would be good to add a baseline that averages all 4 models, possibly with a higher weighting for the area chair model, which is the strongest one. This would help ascertain that the ReFeR approach is doing more than a simple averaging.
>
> The average including AC is a good suggestion, we have shown below the ensemble with weightages including AC with peers:
>
> | Method                | Avg_Spearman | Avg_Kendall-Tau |
> |-----------------------|--------------|-----------------|
> | Llama-3.1-8B          | 0.386        | 0.337           |
> | Mistral-Nemo          | 0.464        | 0.402           |
> | Gemma-2-9B            | 0.563        | 0.489           |
> | G-Eval                | 0.592        | 0.516           |
> | Analyze-Rate          | 0.591        | 0.514           |
> | ReFeR-Lite            | 0.602        | 0.503           |
> | AC with Peer Setup    | 0.5661       | 0.4932          |
> | Peers Avg (without AC model) | 0.539 | 0.447           |
> | Equal Weightage       | 0.5787       | 0.4735          |
> | AC (weight=0.4)       | 0.5864       | 0.4781          |
> | AC (weight=0.5)       | 0.586        | 0.4766          |
> | AC (weight=0.6)       | 0.5866       | 0.4769          |
> | AC (weight=0.7)       | 0.5864       | 0.4768          |
> | AC (weight=0.8)       | 0.5864       | 0.4769          |
> | AC (weight=0.9)       | 0.5857       | 0.4753          |
>
> In the ensemble methods above where AC weightage is given, the remaining weightage (out of 1) will be split equally between the remaining peers, Equal weightage refers to the case where all the Peers and AC model are given equal weightage. As the weightages increase, the performance increases ever so slightly but plateaus later. We can see that this baseline still falls short to some of the other baselines like G-Eval & Analyze-Rate. This shows how simple ensembling like averaging of a higher models’ evaluation might not work.
>
> >3.  I still have a lot of minor comments regarding the terminology, typos, presentations, and potential extensions. These are listed under the "Suggestions / Questions / Typos" section below.
>
> We have addressed all those concerns below.

---

> > ### Author Response · Authors · 2025-06-15
> > **Rebuttal (2/3)**
> >
> > # **Suggestions/Questions/Typos:**
> >
> > > 1. This is a minor terminology point, but I am a little ambivalent about the term "ReFeR-Turbo"......Maybe "Pro" or something like that?
> >
> > Yes, we have changed that in the camera ready, pro is more appropriate.
> >
> > > 2.  Rather than taking n=20 samples from the area chair model (as is the case for ReFeR-Turbo),....... I think using n=1 from 2 different, held-out area chair models could potentially yield better / competitive results than using n=20 from the same area chair model.
> >
> > Thank you for your thoughtful suggestion to use *n=1* from multiple area chair models instead of *n=20* from a single model to address practicality and sample correlation issues. We are attaching the results for the same below, with claude-haiku-3.5
> >
> > | Method                     | Model        | Avg Spearman | Avg Kendall-Tau |
> > |----------------------------|--------------|--------------|-----------------|
> > | ReFeR-Pro (N=20)           | GPT AC       | 0.6352       | 0.5188          |
> > | ReFeR-Lite (N=1)           | GPT AC       | 0.5827       | 0.5041          |
> > | ReFeR-Pro (N=20)           | Claude AC    | 0.6593       | 0.5414          |
> > | ReFeR-Lite (N=1)           | Claude AC    | 0.6325       | 0.5525          |
> > | Averaged ReFeR-Lite (N=1)  | GPT + Claude | 0.6327       | 0.5378          |
> >
> > The GPT AC rows refer to our original setup of GPT-4o-mini as our AreaChair model, and Claude AC refers to the Claude-3.5-haiku model as AC. We can observe that with the suggested method of 2 area chair ReFeR-Lite version, we achieve slightly more than the ReFeR-Lite method, whereas we can see that for both AC, the ReFeR-Pro gives significant gains, compared to the AC. These results demonstrate how the ReFeR method is effective even with multiple Area chairs.
> >
> > > 3. Rather than using manually defined prompts to create the reviewers & area chair model, would using automated prompt discovery methods (e.g. prompt tuning & prefix tuning) and / or parameter-efficient fine-tuning (e.g. LoRA, adapters, etc.) yield better peer reviewers & area chair models?
> >
> > Thank you for the suggestion. While prompt tuning or parameter-efficient finetuning methods like LoRA could potentially improve performance, our goal in ReFeR was to develop a tuning-free, inference-time framework. Instead, we focused on designing a robust manual prompting schema, just like previous works (G-Eval, Analyze-Rate) and introduced auto-generated evaluation guidelines, which significantly improved results even without training.
> > Our experiments show that with just one prompt per evaluation (ReFeR-Lite), we already outperform baselines that rely on 20 sampled responses. But we also did explore prompt optimization techniques like Protegi and have proven how those methods are computationally expensive but don't bring much improvements compared to our schema.
> >
> > > 4. Could the peer reviewers / area chair model generate out-of-range scores (e.g. "6" while the range is 1-5)? If so, what happens in that case?
> >
> > No, the peer and AC models do not give out-of-bound scores when clearly instructed, thanks to their strong instruction-following and reasoning capabilities — which is precisely why we chose them as evaluators. But we still have proper error handling, if an out-of-bound score is detected, we automatically re-run the evaluation. In all our experiments, we never encountered such a case.
> >
> > > 5. How much worse would the result be if ReFeR uses different personas of the same model for the peer reviewers, rather than 3 different models?
> >
> > Chat-Eval, one of our baselines, exactly uses the methodology of different personas with the same model. They also use 3 models, and we replicate their framework, and from the results we can see that the multiple persona framework significantly falls short, even compared to later methods like G-Eval, and Analyze-Rate. Hence, our results show that multiple personas framework scores are a bit lower than other types of frameworks. But given that it would be interesting to see how chateval setup of peers plus our area chair would work. We leave that to future works.

---

> > > ### Author Response · Authors · 2025-06-15
> > > **Rebuttal (3/3)**
> > >
> > > >6.  Typo: On page 6, "... comparable performance which being significantly cheaper". which should be replaced by while.
> > >
> > > Thank you, we corrected the Typo.
> > >
> > > > 7. Would it be possible to ensemble different metrics, e.g. G-Eval & ReFeR Turbo / Lite? It would be great if we can get better results with ensembling, which suggests that the metrics are complementary.
> > >
> > > We ensemble scores from G-Eval and our ReFeR metrics to see if it improves based on the reviewer’s suggestion to see if they are complementary. And these are the results (spearman correlation):
> > >
> > > | Method                          | Coherence | Engagingness | Groundedness | Naturalness | Average |
> > > |--------------------------------|-----------|--------------|--------------|-------------|---------|
> > > | G-Eval (n=20)                  | 0.596     | 0.503        | 0.549        | 0.616       | 0.566   |
> > > | ReFeR-Turbo                    | 0.589     | 0.689        | 0.637        | 0.626       | 0.635   |
> > > | ReFeR-Lite                     | 0.520     | 0.669        | 0.600        | 0.575       | 0.591   |
> > > | G-Eval, ReFeR-Turbo Average    | 0.609     | 0.630        | 0.635        | 0.634       | 0.627   |
> > > | G-Eval, ReFeR-Lite Average     | 0.593     | 0.639        | 0.628        | 0.624       | 0.621   |
> > > | Weighted G-Eval, ReFeR-Turbo Avg | 0.601   | 0.543        | 0.576        | 0.635       | 0.589   |
> > > | Inverse Weighted Avg           | 0.616     | 0.585        | 0.617        | 0.637       | 0.614   |
> > >
> > > We ensemble by taking the average of the scores, in case of G-Eval and ReFeR-Turbo, we average over the 20 + 20 scores, and in case of ReFeR-Lite, we just take the average of the ReFeR-Lite and G-Eval. In case of the weighted G-Eval, ReFeR-Turbo Avg, we have chosen the weights using variance based confidence, basically assuming lower variance over 20 responses meaning higher confidence/weightage (This method is chosen because that is the only external factor to do some kind of ensembling other than simple average). From the results we can see that, for ReFeR-Turbo, this ensemble deteriorates the performance, but for ReFeR-Lite, it slightly improves the performance. In the case of the weighted average, it reduced the performance drastically. But surprisingly in case of the inverse weighted avg, where higher variance was preferred, the performance reductions are not that high.
> > >
> > > >8. Typo: On page 8, "... can result in performance better than the base ....". performance better -> better performance.
> > >
> > > We corrected the typo now.
> > >
> > > >9.  In Table 7, the row below GPT-4o-mini just says "ReFeR". This is ambiguous; do the authors mean "ReFeR-Turbo"?
> > >
> > > Yes, it refers to the Turbo/Pro, we corrected such instances.
> > >
> > > >10.  Would the results be improved if the peer reviewers first do debating, and then revise their scores accordingly? Then the area chair evaluator LLM can take these post-debate peer reviewers' output, and then synthesize & adjudicate those instead of the pre-debate ones.
> > >
> > > We have experimented based on the reviewers suggestion. And the results show that even though there are slight improvements over the normal framework with the debate in 2 metrics, overall, just the hierarchical structure without the Debate is better. We also see that the correlation of peers' scores drops drastically after debate. We have adopted the debate framework from the Multi-Agent Debate framework(Du et al., 2023) in our reasoning baselines, we have a single debate round and then the AC review, as per the reviewer’s suggestion.
> > > Here are the results:
> > >
> > > | Category    | Method              | Coherence | Engagingness | Groundedness | Naturalness | Average |
> > > |-------------|---------------------|-----------|--------------|--------------|-------------|---------|
> > > | **ReFeR**       | Llama-3.1-8B        | 0.312     | 0.451        | 0.395        | 0.330       | 0.372   |
> > > |             | Mistral Nemo        | 0.441     | 0.623        | 0.429        | 0.374       | 0.467   |
> > > |             | Gemma-2-9B          | 0.505     | 0.603        | 0.582        | 0.516       | 0.552   |
> > > |             | ReFeR Pro           | 0.589     | 0.689        | 0.637        | 0.626       | 0.635   |
> > > |             | ReFeR Lite          | 0.537     | 0.630        | 0.583        | 0.565       | 0.579   |
> > > | **Debate + AC** | Llama-3.1-8B        | 0.211     | 0.158        | 0.130        | 0.224       | 0.181   |
> > > |             | Mistral Nemo        | 0.134     | 0.034        | 0.012        | 0.109       | 0.072   |
> > > |             | Gemma-2-9B          | 0.466     | 0.449        | 0.436        | 0.433       | 0.446   |
> > > |             | ReFeR Pro           | 0.603     | 0.634        | 0.650        | 0.586       | 0.618   |
> > > |             | ReFeR Lite          | 0.514     | 0.552        | 0.586        | 0.537       | 0.547   |

---

> > > > ### Comment · Reviewer_SJ6p · 2025-07-06
> > > > **Thank You For The Authors' Response**
> > > >
> > > > I would like to thank the authors for the very thorough authors' response. My concerns have been resolved, and I appreciate the additional experiments that show that the proposed approach outperforms a simple weighted ensemble of the area chair + 3 peer reviewers. I am also pleased to see that the proposed ReFeR method is still effective when using N=1 sample from different area chairs. I also appreciate the additional results on ensembling ReFeR + other metrics, in addition to the results that incorporate debating between the peer reviewers (although I am curious why debate harms performance, which is not obvious to me, given that debating helps in some prior work).

---

### Comment · Action_Editor_tu6t · 2025-06-11
**Discussion period**

Hi all,

I wanted to thank again the authors for their submission and to thank the reviewers for sharing detailed reviews.

As we are now already past the middle of the discussion phase, I would like to encourage the authors to carefully read the reviews, revise the submission where appropriate and submit a response to allow further discussion with the reviewers if needed.

Thank you

---

### Decision · Action_Editor_tu6t · 2025-07-16

**Recommendation:** Accept with minor revision

**Additional Comments:**

Please update the paper according to the discussion during the rebuttal, including all the new results that were added in the comments below and all the clarifications and presentation improvements that were discussed.

**Audience:**

Yes

**Audience Explanation:**

The paper touches multiple timely topics: Automatic LLM evaluation, ensembling LLMs and passing information between different LLMs/ agents, and scaling test time compute for evaluation. The authors introduce a framework that is shown to perform well and share their prompts and detailed ablations that could be beneficial for other researchers working on these topics.

**Claims And Evidence:**

Yes

**Claims Explanation:**

This paper introduces ReFeR, a hierarchical framework for evaluating outputs from generative models, inspired by the academic peer-review process. It uses multiple smaller models as "peer reviewers" and a single, potentially more powerful model as an "area chair" to synthesize feedback and produce a final evaluation for text and images. The design is intuitive, especially to an academic audience, which is an advantage, and the correlation results with human ratings are positive compared to alternative baselines. The multiple ablations and test time experiments are also interesting and support the generalizability of the proposed approach. There are still some counter intuitive results that could benefit further exploration and discussion, like better perfomance with "scores only" version that passes less information from "peers" to the AC agent. But overall the empirical results and amount of ablations and analyses offer interesting insights and support the presented framework.